

# Improved estimation of volcanic $SO_2$ injections from satellite observations and Lagrangian transport simulations: the 2019 Raikoke eruption

Zhongyin Cai[1,2], Sabine Griessbach[2], and Lars Hoffmann[2]

[1]Institute of International Rivers and Eco-security, Yunnan University, Kunming, China
[2]Jülich Supercomputing Centre, Forschungszentrum Jülich, Jülich, Germany

**Correspondence:** Zhongyin Cai (z.cai@fz-juelich.de; czypil@gmail.com)

**Abstract.** Monitoring and modeling of volcanic plumes is important for understanding the impact of volcanic activity on climate and for practical concerns, such as aviation safety or public health. Here, we applied the Lagrangian transport model Massive-Parallel Trajectory Calculations (MPTRAC) to estimate the $SO_2$ injections into the upper troposphere and lower stratosphere by the eruption of the Raikoke volcano (48.29°N, 153.25°E) in June 2019 and its subsequent long-range transport and dispersion. First, we used $SO_2$ observations from the AIRS (Atmospheric Infrared Sounder) and TROPOMI (TROPOspheric Monitoring Instrument) satellite instruments together with a backward trajectory approach to estimate the altitude-resolved $SO_2$ injection time series. Second, we applied a scaling factor to the initial estimate of the $SO_2$ mass and added an exponential decay to simulate the time evolution of the total $SO_2$ mass. By comparing the estimated $SO_2$ mass and the observed mass from TROPOMI, we show that the volcano injected 2.1±0.2 Tg $SO_2$ and the e-folding lifetime of the $SO_2$ was about 13 to 17 days. The reconstructed injection time series are consistent between the AIRS nighttime and the TROPOMI daytime measurements. Further, we compared forward transport simulations that were initialized by AIRS and TROPOMI satellite observations with a constant $SO_2$ injection rate. The results show that the modeled $SO_2$ change, driven by chemical reactions, captures the $SO_2$ mass variations observed by TROPOMI. In addition, the forward simulations reproduce the $SO_2$ distributions in the first ∼10 days after the eruption. However, diffusion in the forward simulations is too strong to capture the internal structure of the $SO_2$ clouds, which is further quantified in the simulation of the compact $SO_2$ cloud from late July to early August. Our study demonstrates the potential of using combined nadir satellite observations and Lagrangian transport simulations to further improve $SO_2$ time- and height-resolved injection estimates of volcanic eruptions.

## 1 Introduction

Injections of trace gases and ash by volcanic eruptions pose significant influences on the Earth's environment. Air pollutants such as sulfur dioxide ($SO_2$), released by volcanic eruptions, can lead to a severe public health hazard and increase excess mortality (Schmidt et al., 2011). In addition, volcanic ash and gases can directly interrupt air flights passing through the volcanic plume and cause long-term damages to airplanes through physical and chemical corrosion, such as the sulfidation due to $SO_2$ (e. g., Grégoire et al., 2018; Prata, 2009). Furthermore, volcanic injections can influence the Earth's climate system



through changes of radiative forcing (e. g., Robock, 2000; Kremser et al., 2016). Explosive volcanic eruptions can inject a
significant amount of $SO_2$ into the stratosphere, and oxidation of the $SO_2$ forms stratospheric sulfate aerosol particles. Due to
the limited potential of dry and wet deposition in the stratosphere and due to the small sedimentation velocities, the sulfate
aerosol particles have long lifetimes on time scales from months to years. In summary, as a precursor of stratospheric sulfate
aerosol and being a good proxy for other volcanic injections such as volcanic ash, monitoring and modeling of the injections
and dispersion of volcanic $SO_2$ can help to better understand the impacts of volcanic eruptions.

Although in-situ observations are available for several well-studied volcanoes (e. g., Whitty et al., 2020), remote sensing
measurements from satellite instruments are more suited to provide long-term records and observations on a global scale. At
present, there are several satellite instruments that can provide $SO_2$ measurements. Among them, the Atmospheric Infrared
Sounder (AIRS) (Aumann et al., 2003; Chahine et al., 2006) aboard the National Aeronautics and Space Administration's
(NASA) Aqua satellite provides measurements of $SO_2$ in the upper troposphere-lower stratosphere (UTLS) region (e. g., Carn
et al., 2005; Prata and Bernardo, 2007; Hoffmann et al., 2014). The AIRS measurements are available since 2002 and have both
day- and nighttime near global coverage. The newly operational TROPOspheric Monitoring Instrument (TROPOMI) aboard
the Sentinel 5 Precursor (S5P) (Veefkind et al., 2012), which is a cooperative undertaking between European Space Agency
(ESA) and the Kingdom of the Netherlands since the late 2017, provides daytime $SO_2$ measurements at an unprecedented
spatial resolution (Theys et al., 2017, 2019) covering also the lower troposphere.

Besides the satellite observations, model simulations can help to characterize volcanic eruptions and provide forecasts of
volcanic plume dispersion. In particular, Lagrangian particle dispersion models (LPDMs), which calculate air parcels trajecto-
ries following the fluid flow, are well suited for simulating complex transport processes (Lin et al., 2012). Widely used LPDMs
include the Flexible Particle (FLEXPART) model (Stohl et al., 2005), the Hybrid Single-Particle Lagrangian Integrated Tra-
jectory (HYSPLIT) model (Draxler and Hess, 1998), the Lagrangian Analysis Tool (Lagranto) (Wernli and Davies, 1997),
the Numerical Atmospheric-dispersion Modeling Environment (NAME) (Jones et al., 2007), and the Stochastic Time-Inverted
Lagrangian Transport model (Lin et al., 2003). A new LPDM, the Massive-Parallel Trajectory Calculations (MPTRAC) model,
was recently developed at the Jülich Supercomputing Centre to take advantage of computing resources on state-of-the-art su-
percomputers (Hoffmann et al., 2016; Liu et al., 2020). The MPTRAC model has been successfully used to reconstruct volcanic
$SO_2$ injections (Heng et al., 2016; Hoffmann et al., 2016) and simulate the long-range transport of volcanic $SO_2$ (Wu et al.,
50 2017, 2018).

When simulating volcanic eruptions, suitable injection parameters, including the location, timing, and injection rate are
needed to initialize the LPDM simulations. Despite the importance for an accurate and reliable transport simulation, how-
ever, obtaining an accurate description of the injection parameters is challenging. Due to limited information regarding the
injection parameters, the simplest assumption is a constant injection over the volcano (e. g., Muser et al., 2020; Kloss et al.,
2021). However, uncertainties in the injection parameters can lead to errors in model simulations and consequently conflicting
conclusions for a single volcanic eruption (Fromm et al., 2014). Complex modeling techniques using inversion algorithms and
data assimilation have been developed to estimate volcanic injections (Eckhardt et al., 2008; Kristiansen et al., 2010; Flemming
and Inness, 2013; Heng et al., 2016). Besides, the injection parameters can also be estimated based on backward trajectories





(Hoffmann et al., 2016; Wu et al., 2017, 2018). The study of Heng et al. (2016) showed that forward transport simulation results using initialization strategies based on inverse modeling and backward trajectory method may have comparable quality. Both, the inverse modeling and backward trajectory methods considered here only give estimates of the altitude distribution and timing of volcanic injections. The $SO_2$ mass of air parcels in the altitude- and time-resolved space is assigned by using a prior assumption on the total mass of $SO_2$ injections, which is usually estimated from satellite products. However, estimates of total $SO_2$ mass can be very different from study to study. For instance, the estimation of total $SO_2$ mass from the 2009 Sarychev eruption varies from 0.8 to 1.5 Tg from different studies (Fromm et al., 2014).

Several limitations may exist when using satellite products to estimate total $SO_2$ mass from volcanic eruptions. Large uncertainties exist during the initial stage of volcanic eruptions. The high $SO_2$ concentration in the early plume leads to saturation effects in satellite observations and subsequently, an underestimation of the total mass. Besides, the co-presence of volcanic ash may also hamper the $SO_2$ mass retrieval at the early stage of an eruption (Yang et al., 2010). Although there is higher confidence after the initial stage, the conversion processes of $SO_2$ to sulfate aerosol starts immediately after injection. Therefore, the $SO_2$ total mass burden observed from satellite at a later stage, when the plume is dispersed and the ash sedimented out, also tends to underestimate the total $SO_2$ injection. In addition, the $SO_2$ is often not injected by the volcano at a single time during the initial stage, which further complicates the estimation of the total injected $SO_2$ mass.

The Raikoke volcano (48.29°N, 153.25°E) in the central Kuril Islands erupted during June 2019, sending a particularly large amount of ash and $SO_2$ into the UTLS (Hedelt et al., 2019; Muser et al., 2020; de Leeuw et al., 2021; Horváth et al., 2021). It was estimated that the 2019 Raikoke eruption injected 1.5±0.2 Tg $SO_2$ into the atmosphere (Global Volcanism Program, 2019; Muser et al., 2020; de Leeuw et al., 2021), making it the largest $SO_2$ injection into the UTLS since the 2011 Nabro eruption and the first large volcanic eruption since begin of operations of TROPOMI. Interestingly enough, the Raikoke eruption formed unique features of compact $SO_2$ clouds with confined shapes and sizes ($\sim$300 km in diameter) during the transport and dispersion of the $SO_2$ injections (Chouza et al., 2020; Gorkavyi et al., 2021). Therefore, the 2019 Raikoke eruption provides an ideal test case to assess the ability to reconstruct the injection parameters using the state-of-the-art TROPOMI satellite observations and to test how the reconstruction compares with observations using the older AIRS instrument. In addition, the compact $SO_2$ cloud phenomenon related to the Raikoke eruption provides a unique opportunity to test the simulation of the transport and dispersion of the volcanic $SO_2$. In this study, both questions are being addressed.

The paper is structured as follows. In Sect. 2, we describe the satellite observations of AIRS and TROPOMI and the MPTRAC model as well as our method of reconstructing the injection parameters. The reconstructed injection parameters are presented in Sect. 3.1. In Sect. 3.2, we assess the performance of the MPTRAC model in simulating the transport and dispersion of the injected $SO_2$ in terms of the total mass of the volcanic $SO_2$, the spatial distribution of the $SO_2$ cloud, and the degree of dispersion of the compact $SO_2$ cloud. Finally, we discuss the results from our work comparing with previous studies in Sect. 4 and main conclusions are drawn in Sect. 5.





## 2 Data and methods

### 2.1 AIRS SO$_2$ observations

To estimate the injection parameters of volcanic SO$_2$ and to initialize and validate the forward simulations with MPTRAC, we used SO$_2$ observations from AIRS and TROPOMI. Since May 2002, AIRS/Aqua operates on a polar sun-synchronous orbit
with equatorial crossing time at 01:30 local time for the descending orbit, and at 13:30 local time for the ascending orbit. The scan for each swath covers a width of 1780 km, consisting of 90 footprints, and the along-track distance of two adjacent swaths is 18 km. The sizes of the footprints are 13.5 km $\times$ 13.5 km at nadir and 41 km $\times$ 21.4 km at the scan extremes.

AIRS measures thermal infrared spectra in three bands between 3.74 and 15.4 $\mu$m. For the SO$_2$ detection, we used the SO$_2$ index (SI) defined by Hoffmann et al. (2014), which identifies the brightness temperature difference (BTD) between two
different radiance channels (1407.2 and 1371.5 cm$^{-1}$) from the AIRS spectral measurements in the 7.3 $\mu$m SO$_2$ waveband. The SI provides SO$_2$ information for the atmospheric column, but no vertical profile is directly available. The kernel function (Fig. 1) for the SI, based on radiative transfer calculations for a mid-latitude atmosphere (Hoffmann et al., 2014), shows that the SI is most sensitive to SO$_2$ layers at 8 to 13 km. The SI is measured in units of Kelvin and increases with increasing SO$_2$ column density. Here, we used a correlation function derived from the radiative transfer calculations of Hoffmann et al.
(2014) for a mid-latitude atmosphere to convert the SI to SO$_2$ column density. Based on our inspection of the AIRS data, measurements beyond a threshold of 1.4 K or 5 DU are clearly indicating the presence of volcanic SO$_2$ from the 2019 Raikoke eruption. However, due to the conversion using an approximate correlation function, our estimates of total SO$_2$ mass from AIRS are generally considered to be less reliable and total SO$_2$ mass will rather be obtained from the TROPOMI products in this study.

As an example, Fig. 2 shows plots of the Raikoke SO$_2$ clouds on 26 June 2019 as retrieved from AIRS (nighttime and daytime data, respectively) and TROPOMI (daytime data, only) measurements. Besides differences caused by the $\sim$12 hour time shift, we found that the AIRS nighttime and daytime observations were not always consistent with each other. They also showed some differences when reconstructing the Raikoke injection parameters. The AIRS daytime measurements are possibly influenced by scattering of solar radiation at the surface or at cloud or aerosol layers at upper levels. Therefore,
the AIRS nighttime and daytime data are considered separately in this study and daytime data have been treated particularly carefully.

### 2.2 TROPOMI SO$_2$ observations

The TROPOspheric Monitoring Instrument (TROPOMI) is a single instrument on ESA's Copernicus Sentinel-5 Precursor satellite that was launched in October 2017. Sentinel-5P's mean local solar time of the ascending node is 13:30 and its orbit
is aligned with NASA's Suomi-NPP mission (approximately 5 minutes behind) to allow for synergistic use with Suomi-NPPs cloud products (Veefkind et al., 2012). TROPOMI consists of four passive grating imaging spectrometers measuring in the UV, VIS, NIR, and SWIR (Veefkind et al., 2012) and hence, provides daytime measurements only. TROPOMI is a nadir instrument





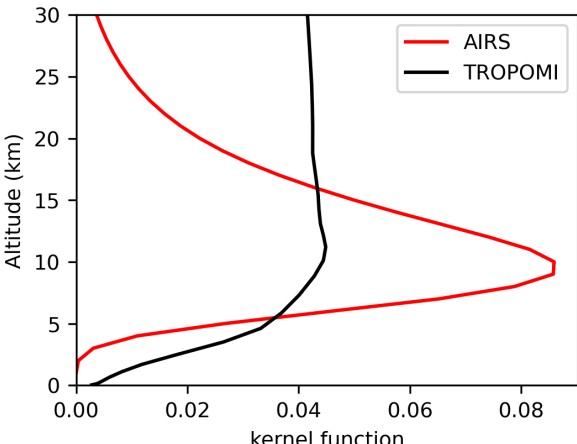

**Figure 1.** Representative kernel functions for AIRS $SO_2$ observations at mid latitudes and for TROPOMI $SO_2$ observations over the Raikoke region.

with a swath width of $2600\,\text{km}$ and a very high spatial resolution of $7 \times 3.5\,\text{km}^2$ (until August 2019) (Veefkind et al., 2012; Romahn et al., 2021).

For the analysis of the Raikoke eruption we used the TROPOMI Level 2 offline (OFFL) V01.01.07 $SO_2$ data product for the time period between 20 June 2019 and 16 August 2019. The TROPOMI $SO_2$ data product provides four total vertical columns of $SO_2$ in $\text{mol}\,\text{m}^{-2}$, one for the total atmospheric column between the surface and the top of the atmosphere and three columns assuming an $SO_2$ layer at 1, 7, and 15 km altitude in the retrieval. The details of the retrieval are given in Theys et al. (2017, 2021). In studies investigating volcanic plumes, it is common to use the vertical column densities retrieved for distinct plume

heights (e. g., Theys et al., 2019). In this study, we used the total vertical $SO_2$ column of the 15 km retrieval, as we considered it to provide the best approximation for the Raikoke eruption as in other studies (e. g., Muser et al., 2020; de Leeuw et al., 2021). Compared with AIRS, the lower detection limit for TROPOMI is 0.3 DU (Theys et al., 2021) and data below this threshold are discarded in this study.

## 2.3   The MPTRAC model

Massive-Parallel Trajectory Calculations (MPTRAC) is a Lagrangian particle dispersion model for the analysis of atmospheric transport processes in the troposphere and stratosphere (Hoffmann et al., 2016). It calculates particle trajectories by solving the kinematic equation of motion using given wind fields from reanalysis or forecast meteorological data. The MPTRAC model currently uses the midpoint method to solve the equation of motion, which gives the optimized balance between accuracy and computational efficiency (Rößler et al., 2018). Besides vertical motion driven by the vertical velocity (i. e., kinematic

trajectories), the MPTRAC model provides options to constrain the pressure of the air parcels to either constant pressure (isobaric surface), constant density (isopycnic surface), potential temperature (isentropic surface), or pressure time series from





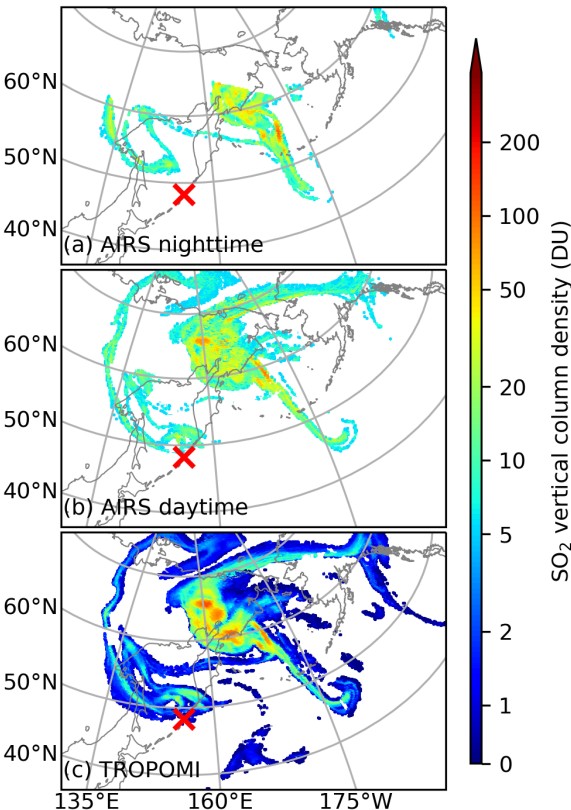

**Figure 2.** Spatial distribution of SO$_2$ vertical column density (DU) during the 24 hour period between 26 June 2019, 12:00 UTC and 27 June 2019, 12:00 UTC from AIRS nighttime (a), AIRS daytime (b), and TROPOMI daytime (c) observations. Note that AIRS observations of SO$_2$ vertical column density less than 5 DU are not shown here as those data are not actually used in the analysis because they are affected by background noise. The AIRS nighttime observations have a ∼12 hour time shift compared with the AIRS daytime and TROPOMI observations.

balloon measurements (Hoffmann et al., 2017). In addition, the model also includes turbulent diffusion and subgrid scale wind fluctuations to simulate the diffusion. The turbulent diffusion is described by fixed diffusivity coefficients. Following the FLEXPART model (Stohl et al., 2005), the MPTRAC model uses a constant horizontal diffusivity of $50\,\mathrm{m^2s^{-1}}$ for the troposphere and a vertical diffusivity of $0.1\,\mathrm{m^2s^{-1}}$ for the stratosphere as default values. Subgrid scale wind fluctuations are simulated using the Langevin equation to add time-correlated stochastic perturbations to the trajectories. The subgrid scale wind standard deviations are downscaled from the grid scale standard deviations by using a default scaling factor of 40 % (Stohl et al., 2005). To investigate the effect of parameterizations of turbulent diffusion and subgrid scale wind fluctuations on the simulated SO$_2$ dispersion for the Raikoke case, we varied the diffusivity and the scaling factor of the subgrid variance separately. As the actual diffusivity can vary by several orders of magnitude (e. g., Ishikawa, 1995; Desiato et al., 1998; Legras



et al., 2005; Pisso et al., 2009), we tested the turbulent diffusion by varying the diffusivities from $10^{-2}$ to $10^3$ times the default values. For subgrid scale wind fluctuations, we varied the scaling factor from 0 to 100 %.

Additional modules are implemented to simulate convection, sedimentation, radioactive decay, hydroxyl chemistry, dry deposition, and wet deposition. In this study, we used the hydroxyl chemistry module to simulate the loss of $SO_2$ by its reaction
with the hydroxyl radical (OH). The MPTRAC model also provides variable output methods. In this study, we implemented a new module for "sample output", which allows us to sample the model data at the exact time and location of the satellite overpasses/footprints. In addition to the trajectory and gridded outputs, the model also provides ways to directly evaluate the performance of the simulations, such as calculating the critical success index (CSI) (Wilks, 2011). Basically, the CSI is based on the counts of detection by the observation and simulation on a regular grid basis. If the vertical column density in a grid cell
passed a user specified threshold, it will be counted as "yes", otherwise it will be counted as "no". The CSI is the ratio between the count of hits and the total number of hits, false alarms, and misses. Along with CSI, the probability of detection (POD) and the false alarm rate (FAR) are also calculated.

In this study, the main meteorological data used to drive the MPTRAC simulations is taken from the ERA5 reanalysis. The ERA5 is the ECMWF's (European Centre for Medium-Range Weather Forecasts) fifth generation reanalysis (Hersbach
et al., 2020), which is meant to replace its predecessor ERA-Interim (Dee et al., 2011). ERA5 provides hourly outputs of a comprehensive set of variables at 31 km horizontal resolution and 137 levels spanning from the surface up to 0.01 hPa. In this study, the ERA5 data are interpolated to $0.3° \times 0.3°$ horizontal resolution. In comparison, the ERA-Interim data have a horizontal resolution of 80 km, 60 model levels, and output every 6 hours, i. e., at 00:00, 06:00, 12:00, and 18:00 UTC. The differences between ERA5 and ERA-Interim in driving Lagrangian transport simulations have been assessed by Hoffmann
et al. (2019), finding that the choice of data has a considerable impact on the simulations, in particular due to better spatial and temporal resolutions of the ERA5 data. We also considered both, ERA5 and ERA-Interim data in this study, with a major focus on results derived from the ERA5 reanalysis.

## 2.4 Estimation of volcanic $SO_2$ injections

To reconstruct the altitude- and time-resolved injection parameters of the Raikoke eruption, being represented by the altitude,
time, and $SO_2$ mass of each air parcel over the volcano, we used a method based on backward trajectories released from the columns of the AIRS and TROPOMI $SO_2$ measurements (Hoffmann et al., 2016; Wu et al., 2017, 2018). The analysis was done separately for AIRS and TROPOMI data, covering time periods from a few days up to weeks after the eruption, and the results were compared against each other. As both, AIRS and TROPOMI provide information on the horizontal location and time of the $SO_2$ observations, but lack information on the vertical distribution of the $SO_2$, we released multiple air parcels
between 0 and 25 km altitude at each individual satellite footprint with volcanic $SO_2$ detections. In contrast to our earlier work, the vertical profile of the number of air parcels has been made to follow the mean kernel function of the satellite measurements (Fig. 1) in order to take into account their different vertical sensitivity. The total number of air parcels at each location was linearly proportional to the total column density of the satellite observations. At the same time, a Gaussian scatter of the air





parcels with 15 and 5 km full width at half maximum (FWHM) was introduced to represent the horizontal footprint size for
AIRS and TROPOMI, respectively.

In total, 5 million air parcels were released to calculate backward trajectories. If a backward trajectory passed the Raikoke
volcano within a search radius of 15 km, the location and time of the air parcel was saved to reconstruct the injection parameters.
We note that, based on our sensitivity tests, the results are not very sensitive to FWHM (i. e., between 1 and 50 km) and the
search radius around the volcano (i. e., between 1 and 100 km). All backward trajectories that met the selection condition were
re-sampled to a total number of 5 million particles and an initial total mass of 1.5 Tg was assigned to them. After the initial
relative $SO_2$ distribution has been estimated from the backward trajectories, we conducted forward simulations and applied a
scaling factor to the $SO_2$ total mass for further calibration. To calibrate the total mass, we assumed that the $SO_2$ starts to decay
exponentially with a fixed e-folding lifetime immediately after the injection and compared the change of the $SO_2$ total mass
from the simulations with the change of the $SO_2$ total mass derived from TROPOMI observations.

## 3   Results

### 3.1   Volcanic $SO_2$ injections parameters

#### 3.1.1   Final reconstruction of Raikoke $SO_2$ injections

Figure 3a shows the final reconstruction of the Raikoke $SO_2$ injections based on TROPOMI observations and Lagrangian
transport simulations using ERA5 winds. The mass in the reconstruction has been turned to achieve a total injection of 2.1 Tg.
The altitude- and time-resolved injection, and the integrated vertical profile are also shown in Fig. 3. A major $SO_2$ emission was
reconstructed during the first two days of the time series, i. e., between 21-22 June 2019. After this major eruption, significantly
smaller amounts of $SO_2$ were continuously injected by the volcano until the end of June with a prominent second and third
plume during 24-25 June and 27-28 June, respectively. The first plume crossed the tropopause and injected $SO_2$ between 5 and
15 km of altitude, with ∼45 percent of the $SO_2$ mass reaching the stratosphere (Fig. 3a and b). The second and the third plumes
205   mainly injected material into the troposphere. As the Raikoke eruption is dominated by the first plume, the overall injected
$SO_2$ (Fig 3c) distributes around the tropopause with peak injections at an altitude of 11 km.

As an intercomparison as well as a validation, the vertical profiles integrated over the entire eruption period (21 to 30 June
2019) of our different injection estimations and the profile derived by the VolRes team (de Leeuw et al., 2021) are shown
together in Fig. 4. The profile derived by the VolRes team is mainly based on IASI observations during the first two days
210   of the Raikoke eruption (de Leeuw et al., 2021). Compared with the VolRes profile, the altitude of peak injections is about
1 km above the VolRes profile, no matter which satellite data, i. e., AIRS nighttime or TROPOMI daytime, and reanalysis data,
i. e., ERA5 or ERA-Interim, have been used. Our reconstructions also show enhanced injections between 12 and 14 km, being
consistent with de Leeuw et al. (2021) that the injections reached higher into the stratosphere than indicated by the VolRes
estimation. When excluding the second and third plume, the vertical profile for the first major eruption (figure not shown) is
similar with the overall injection profile (Fig. 4) with slightly reduced emission rate in the stratosphere and larger reduction





**Figure 3.** Reconstructed $SO_2$ injections of the 2019 Raikoke eruption based on TROPOMI observations. (a): altitude-resolved $SO_2$ injection rate time series $(kg/(m\,s))$. (b) temporal evolution of the vertically integrated $SO_2$ injection rates $(kg/s)$ for the whole atmosphere, the troposphere, and the stratosphere. The temporal change of accumulated $SO_2$ injections (integrated for the whole atmospheric column and over time) is also plotted in (b). (c): altitude profile of the $SO_2$ injection rates $(kg/m)$.

in the troposphere part. The VolRes profile also indicates a small peak at low altitudes around 2 km. However, this part is not found in our reconstruction. The most likely reason is that both, AIRS and TROPOMI, have a limited sensitivity in the lower troposphere, i.e., below 5 km (Fig. 1).





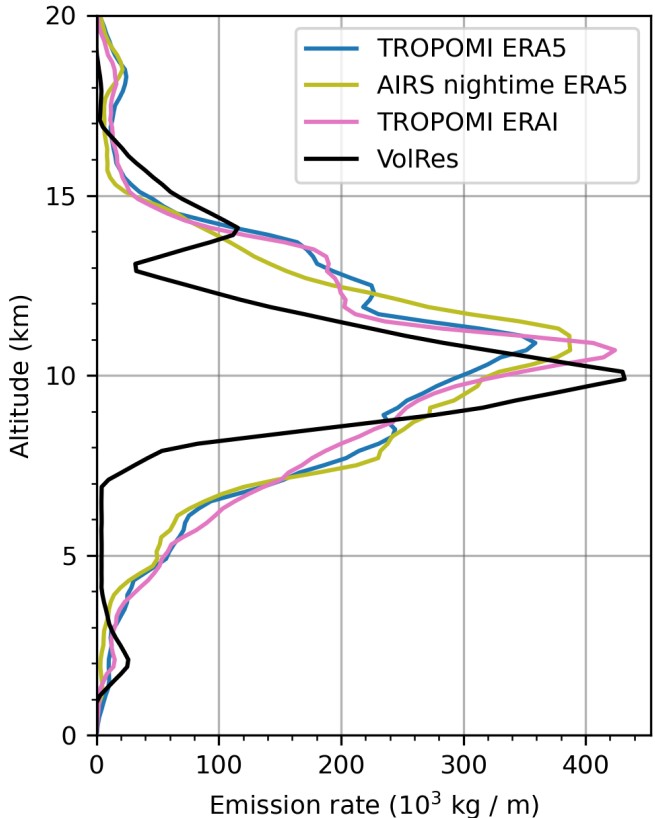

**Figure 4.** Vertical profiles of Raikoke $SO_2$ injections derived by the VolRes team and from different combinations of datasets (see plot key).

### 3.1.2 Calibration of the total mass of the $SO_2$ injections

For the initial reconstruction, we estimated the $SO_2$ injection rates under the assumption of a $SO_2$ total mass of 1.5 Tg, as found by the VolRes team. Figure 5 shows the time series of the vertically integrated $SO_2$ injections. The comparison of the reconstructions based on different satellite observations (Figure 5) show that the results derived from AIRS nighttime and TROPOMI observations agree well. The reconstruction derived from the AIRS daytime observations shows weaker injections in the first plume, but the second and third plume are stronger compared to the reconstructions based on AIRS nighttime and

TROPOMI measurements. As pointed out in Sect. 2.1, we will focus our analyses on the AIRS nighttime and TROPOMI results in the following parts.

To estimate the final total injected $SO_2$ mass, we derived the daily $SO_2$ mass from the TROPOMI observations (Fig. 6). The TROPOMI observations shows that a total $SO_2$ mass of ∼1.4 Tg peaked during 24-26 June, while the cumulative $SO_2$ injection from the initial reconstruction at 26 June is only 1.2 Tg. When the cumulative $SO_2$ injection in the initial reconstruction reached

1.5 Tg, the total $SO_2$ mass from TROPOMI decreased to 1.2 Tg due to the removal of $SO_2$. To better represent the evolution





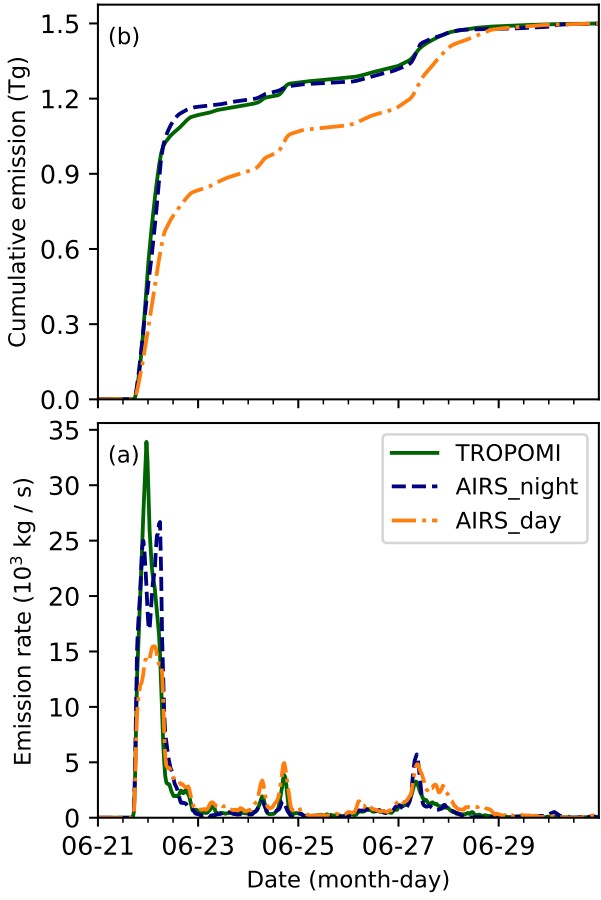

**Figure 5.** Temporal change of Raikoke SO$_2$ injections reconstructed based on TROPOMI observations (green line) and AIRS observations during nighttime (blue dashed line) and daytime (orange dash-dotted line): (a) vertically integrated SO$_2$ injection rate, and (b) accumulated SO$_2$ mass.

of the total SO$_2$ mass in the simulations, we scaled our initial mass reconstruction to the TROPOMI data and applied an exponential decay to account for the removal of SO$_2$ (Fig. 6). In this experiment, we found that a total injection of 1.9 to 2.3 Tg SO$_2$ and an e-folding lifetime of 13 to 17 days best represents the temporal evolution of total SO$_2$ mass in the atmosphere. Therefore, we re-scaled the initial reconstruction to a total mass of 2.1 Tg. Note that although the e-folding lifetime of 13-17

days well represents the overall removal of SO$_2$ injections for the Raikoke case, SO$_2$ removal rates in general are very sensitive to the altitude of the SO$_2$ injections and the atmospheric background conditions.





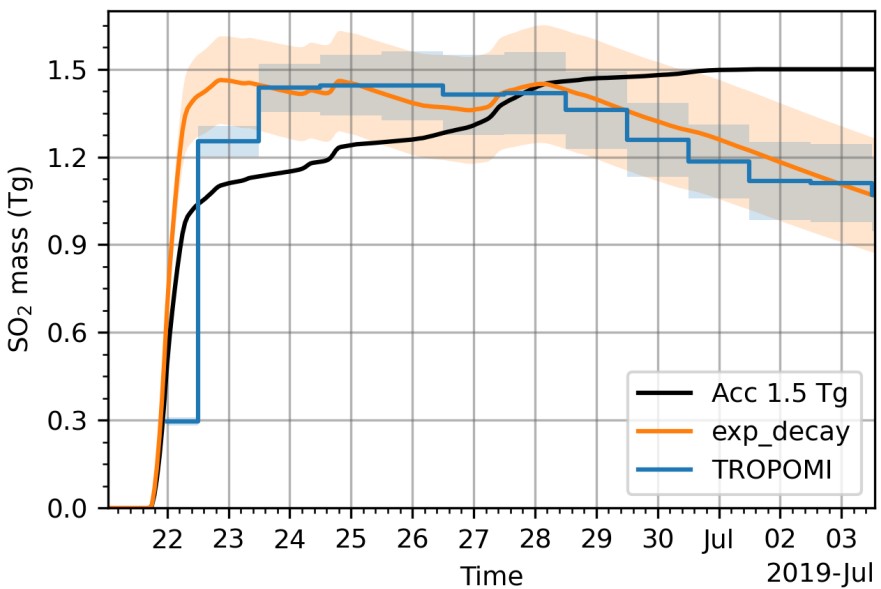

**Figure 6.** Temporal change of total $SO_2$ mass from TROPOMI measurements (blue), and calculated total mass with an e-folding lifetime of 15 days (orange) and a mass scaling factor of 2.1/1.5. Orange shadings show the combination of the scaling factor ranging between 1.9/1.5 and 2.3/1.5 and e-folding lifetime ranging between 13 and 17 days. The black curve shows the accumulated $SO_2$ injection with a total injection of 1.5 Tg (the initial reconstruction).

### 3.1.3 Sensitivities of estimated $SO_2$ injections on data and model parameters

Investigating the sensitivity of the reconstructed injection time series on the underlying input data, we ran the reconstruction using AIRS $SO_2$ measurements together with ERA5 winds as well as TROPOMI measurements together with ERA-Interim winds. In comparison to TROPOMI and ERA5, the overall patterns of injection estimations based on AIRS nighttime observations and ERA5 (Fig. 7a) or TROPOMI observations and ERA-Interim (Fig. 7b) are quite similar. For the first plume, i. e., between 21 June and 22 June, the estimation based on AIRS nighttime observations and ERA5 shows stronger injections during the beginning and late stage of the plume, while the estimation based on TROPOMI observation and ERA-Interim shows weaker (stronger) injections at the beginning (late) stage of the first plume, respectively. Differences do exist during the second and the third plumes, but they are relatively small.

We conducted more than two hundred simulations to test the sensitivity on the injection reconstructions. Among the tested parameters, we found that the coverage of the satellite observations has the largest impact on the injection reconstruction. More specifically, it matters how many days of satellite observations are used for the reconstruction and how close to the location of the volcano satellite observations are discarded. Here, we only describe the sensitivity tests on the temporal and spatial coverage of the satellite observations, whereas the sensitivity tests on other parameters are not shown.





**Figure 7.** Differences of reconstructed SO$_2$ injections of the 2019 Raikoke eruption derived from different combinations of satellite observations and reanalysis data compared with the TROPOMI-ERA5 combination: AIRS nighttime with ERA5 (a), and TROPIMI with ERA-Interim (b).

Figure 8 shows the SO$_2$ mass change in forward simulations initialized by using TROPOMI observations covering different numbers of days since the beginning of the eruption. The total SO$_2$ mass in all the simulations was assigned to 2.1 Tg. As shown in Fig. 8, when using just a few days of observations, the simulation produces a too strong peak at the beginning of the volcanic eruption. Increasing the time period of the satellite data, a gradual decrease of the first peak and redistribution of SO$_2$
to a later stage of the eruption is observed. Therefore, using short term observations will lead to a more pronounced first plume, and on the contrary, longer term observations will increase the amplitude of the second and third plume. The sensitivity test shows that using 12 days of observations gives an optimal representation of the SO$_2$ mass.





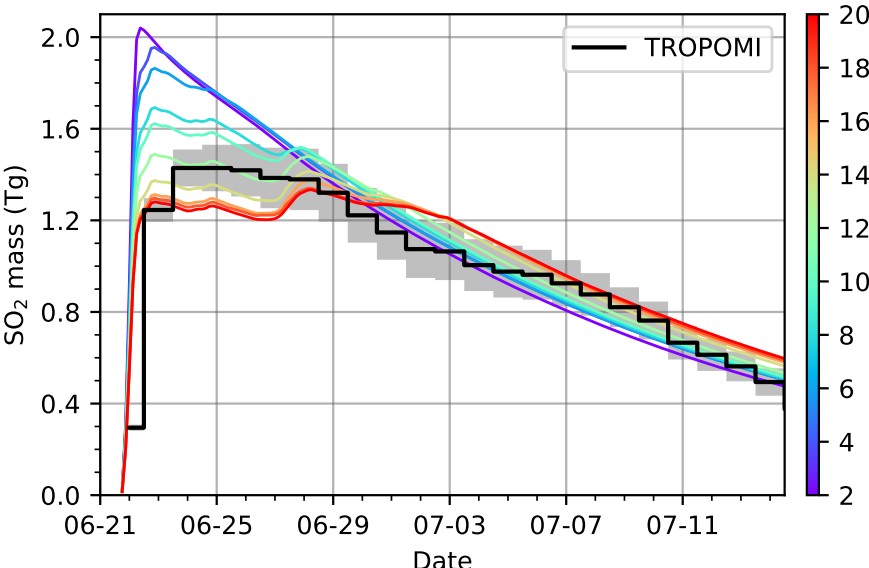

**Figure 8.** Temporal change of $SO_2$ total mass in MPTRAC forward simulations which were initialized by considering different numbers of days of TROPOMI observations since the beginning of the Raikoke eruption (see color bar). Total mass of $SO_2$ injection in all simulation is 2.1 Tg. The mass changes measured by TROPOMI are shown by the black line and gray shading indicates the measurement errors.

As the backward trajectory method heavily relies on the quality of the trajectories, satellite observations too close to the volcano can not provide enough information to separate between different altitudes. Therefore, we defined a circle with a
certain distance to the Raikoke volcano, were the satellite observations falling inside the circle are discarded. When using TROPOMI observations and the distance being set to a very small value, such as a few kilometers, most of the reconstructed $SO_2$ is injected at the beginning of the eruption. When the distance is being set to several hundred kilometers, similar to increasing the temporal duration of the trajectories, the injection of $SO_2$ at the beginning of the eruption is weakened and more $SO_2$ is injected within a few to several days after the beginning of the eruption. When using the AIRS measurements, this
effect becomes less pronounced. Overall, the AIRS and TROPOMI reconstructions agreed better with each other when setting a larger distance. In the final reconstruction, we used a distance of 750 km, which gave the most consistent results.



### 3.2 Forward simulations for the Raikoke eruption

#### 3.2.1 Simulations of SO$_2$ total mass

We performed forward simulations initialized by the different reconstructed SO$_2$ injection parameters as well as a constant
SO$_2$ injection rate. For the forward simulation with a constant injection, we uniformly assigned 1.5 Tg SO$_2$, which is the initial estimate of the total SO$_2$ injection, from 5 to 15 km and from 21 June 2019, 18:00 UTC to 22 June 2019, 06:00 UTC. Unless noted differently, all forward simulations that were initialized by a constant injection rate in the following sections have the same setup as described here. In the following subsections, we will present and compare the forward simulations of SO$_2$ in terms of total mass burden and spatial distributions. In addition, we also performed different forward simulations driven
by ERA5 and ERA-Interim data. However, the overall patterns of simulated SO$_2$ were generally similar between ERA5 and ERA-Interim. Therefore, if not specified otherwise, the forward simulations driven by the ERA5 data are shown.

In the most recent version of the MPTRAC model, a hydroxyl (OH) chemistry module has been implemented to simulate the removal of SO$_2$ due to chemical reaction with OH. This module enables the direct comparison of total SO$_2$ mass change in model simulations with the simple exponential decay experiments and the satellite observations by TROPOMI (Fig. 9).
In the forward simulations, we have used different injection parameters with total injected SO$_2$ mass ranging from 1.9 to 2.3 Tg. As shown in Fig. 9, the SO$_2$ mass in the forward simulation initialized with a 2.1 Tg total injection, either using injection parameters derived from TROPOMI daytime (Fig. 9a) or AIRS nighttime (Fig. 9b) observations, agrees well with the exponential decay experiment of a 14-day e-folding lifetime. In addition, all the experiments are consistent with the total SO$_2$ mass derived from TROPOMI observations. Figure 9 also shows the SO$_2$ mass change in the forward simulation initialized by
a constant injection time series. In contrast to the forward simulation initialized by our reconstructed injection time series, the simulation initialized with a constant injection rate produced an SO$_2$ mass peak, which is comparable with the maximum SO$_2$ mass in TROPOMI observations, at the beginning of the eruption. Then, the gradual removal of SO$_2$ leads to lower mass in the model simulation than observed by TROPOMI (Fig. 9). From these comparisons, we conclude that the June 2019 Raikoke eruption produced a total injection of 2.1 Tg SO$_2$, which has an overall e-folding lifetime of 14 days in the UTLS region during
the first two weeks after the eruption.

The comparison of the temporal changes of the SO$_2$ total mass among the different forward simulation settings and TROPOMI observations (Fig. 9) suggests that our estimation of 2.1 Tg SO$_2$ injection is reasonable. The initial estimation of 1.5 Tg mainly reflects SO$_2$ injections of the major eruption during the first two days. Consistent with this estimate, the total mass in our estimation for the first plume is about 1.5 Tg (Fig. 3b). However, additional injections after the first plume are required to
reproduce the observed SO$_2$ mass in the model simulations (Fig. 9).

Although an e-folding lifetime of 14 days well captures the overall mass reduction of injected SO$_2$ in the atmosphere, the real removal rates of SO$_2$ at different altitudes are different. Figure 10 shows the remaining mass of SO$_2$ injected to 1 km thick layers during the first 12 hours of the eruption (21 June 2019, 18:00 UTC to 22 June 2019, 06:00 UTC). In general, the removal rate decreases with altitude, mostly because of lower OH concentrations in the lower stratosphere compared to the troposphere.
In the troposphere, the SO$_2$ mass is reduced to less than 50 % within several days to a week, while the stratospheric injections



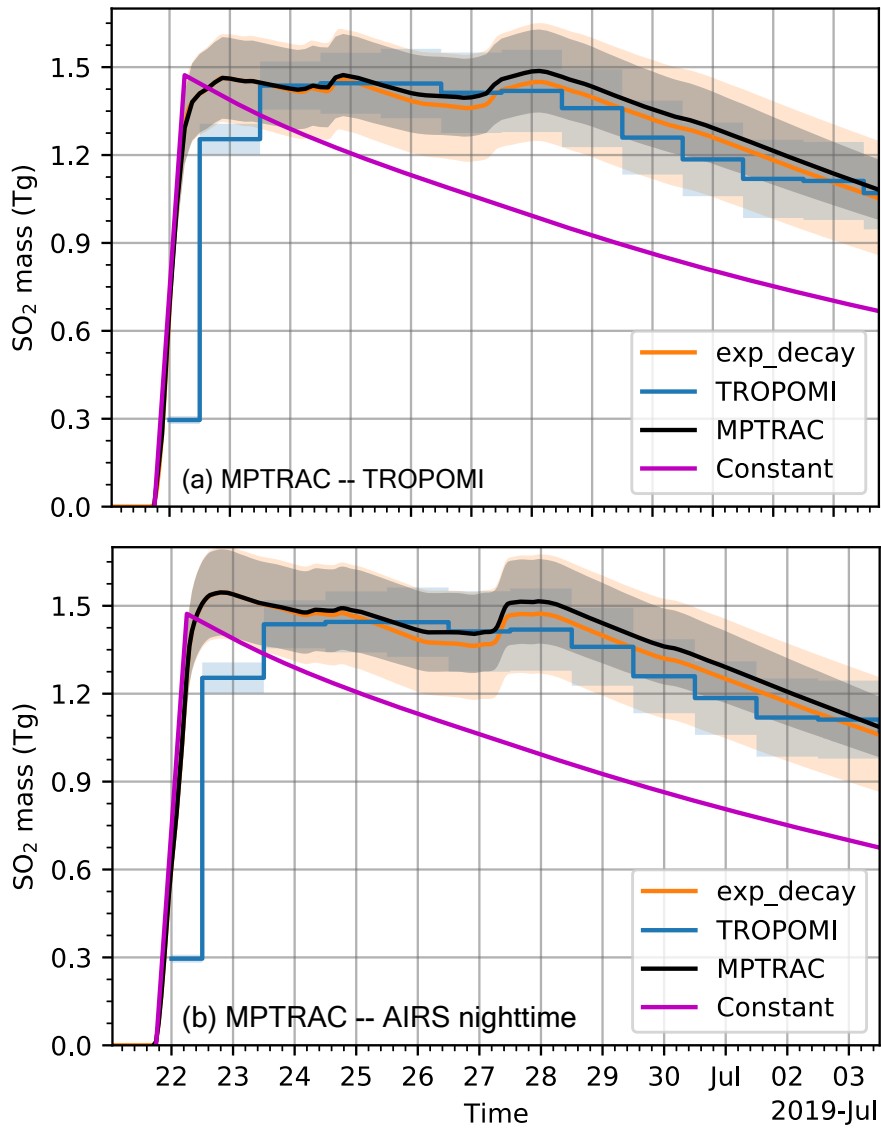

**Figure 9.** Temporal change of total SO$_2$ mass in the MPTRAC forward simulation (black lines) initialized by TROPOMI observations (a) and AIRS nighttime observations (b), respectively. Gray shadings show the range of total injection between 1.9 and 2.3 Tg. The mass changes measured by TROPOMI (blue) and modeled by an exponential decay (orange) from Fig. 6 are repeated here for comparison.

still have ∼70 % at 10 days after the eruption. This means that the tropospheric injections are removed quickly during the early stage of the eruption and the stratospheric injections gradually dominate. In the satellite SO$_2$ observations, the vertical column density of the SO$_2$ cloud associated with the tropospheric injection also decreases faster than the SO$_2$ cloud associated with





stratospheric injection (Fig. 11; see details below). This observation also confirms the faster removal of tropospheric parts of
the SO$_2$ injections.

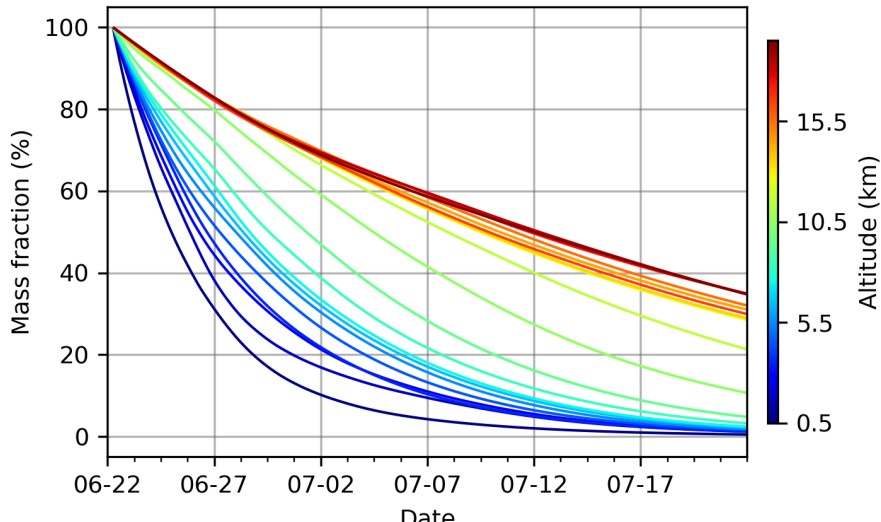

**Figure 10.** Temporal change of the remaining fraction of SO$_2$ mass for injections at different 1 km thick layers between 21 June 2019, 18:00
UTC and 22 June 2019, 06:00 UTC. Colors indicate the altitude at the middle of each 1 km thick layer.



### 3.2.2 Simulation of SO$_2$ transport during the first ∼10 days

In general, the forward simulations initialized by both the TROPOMI daytime and AIRS nighttime observations well reproduce the spatial distribution of SO$_2$ during the first ∼10 days of simulation time, especially in terms of spatial location and extent. We note that the forward simulations initialized by the TROPOMI and AIRS nighttime observations are highly consistent with each other, as the injection parameters estimated from these two datasets do not differ fundamentally (Fig. 7). Therefore, the results from the forward simulation initialized by the AIRS nighttime observations are not shown here. To illustrate the performance after major SO$_2$ injections of Raikoke, we selected three satellite overpasses on 23 June, 25 June, and 28 June to show the SO$_2$ distribution in observations and model simulations (Fig. 11).

The TROPOMI observations and the MPTRAC simulations show that the SO$_2$ injections separated into two major clouds (Fig. 11). We added mean trajectories for injections between 7-8 km and 11-15 km in Fig. 11 to indicate the major movements of the two clouds. Both of the clouds are moving cyclonically. A smaller SO$_2$ cloud, which is represented by the mean trajectory for injections between 7-8 km, moves faster and the SO$_2$ column density also decreased very fast to less than 10 DU on 28 June. The SO$_2$ column density in the other major cloud, which mainly reflects the stratospheric injections, decreased much slower compared with the SO$_2$ cloud that reflects tropospheric injections. This observation by TROPOMI is consistent with the faster removal of SO$_2$ in the troposphere (Fig. 10).

After the first injection, TROPOMI observations show that the SO$_2$ clouds are located to the east of the Raikoke volcano, but split into two branches with one branch being in the north and the other branch being in the south (Fig. 11a). The forward simulations initialized by the TROPOMI observation and a constant injection rate reproduce the main northern branch, which locates just to the east of the Raikoke (Fig. 11b and 11c). However, both simulations only reproduce a part of the southern branch and the part reproduced in the simulation initialized by a constant injection rate is too strong. Comparing with the major northern branch SO$_2$ cloud, note that the southern branch is very weak with SO$_2$ column densities mostly less than 10 DU. A sensitivity test suggested that the southern branch is mainly associated with transport of SO$_2$ in the lower troposphere (between 0 and 5 km), which was not represented in both initializations.

After the second plume by 25 June (Fig. 11d – f), most of the SO$_2$ injections were moved to the northwest direction over the Asian continent and to the northeast direction over the northwest Pacific ocean (Fig. 11d). In addition to these two major SO$_2$ clouds, there is a weaker SO$_2$ cloud to the east of the Raikoke volcano, which is probably related to the injections between 23-25 June (Fig. 3). The forward simulation initialized by TROPOMI observations reproduced the general pattern of the three clusters. However, the forward simulation initialized by a constant injection rate only reproduces the two major SO$_2$ clouds in the northwest and the northeast directions.

After the third plume by 28 June (Fig. 11g), the SO$_2$ cloud that was over the Asian continent in the northwest direction now moved back to the east of Raikoke along a cyclonic circulation. In contrast, the SO$_2$ cloud that was over the northwest Pacific ocean showed a slower movement and is now located over the Asian continent (Fig. 11g). Both forward simulations reproduced the two major SO$_2$ clouds. The forward simulation initialized by the TROPOMI observations reproduced a stronger SO$_2$ cloud to the east of the Raikoke volcano (Fig. 11h) due to injections during the third plume. Similarly, the simulated SO$_2$





cloud over the volcano after the second plume is also stronger than in the observations (Fig. 11e). This result indicates that
our reconstructed injection parameters potentially overestimate the second and the third plume. However, totally removing the
second and/or the third plume would severely reduce the ability to correctly simulate the total $SO_2$ mass burden as shown in
Fig. 9.

Although the forward simulations can reproduce the observed $SO_2$ distributions with relatively high performance during
the first week (Fig. 11), the simulation starts gradually losing the ability to capture the structures of the $SO_2$ cloud thereafter.
Fig. 12 shows the observed and simulated spatial distribution of the $SO_2$ at the beginning of 1 July. Overall, the $SO_2$ distributes
like a strip pattern with major peaks over the Sea of Okhotsk and the west coast of the Bering Sea (Fig. 12a). The model
simulation captured this overall strip like pattern and even some fine details over northern high latitudes (Fig. 12b). However,
the simulated $SO_2$ distribution does not correctly reproduce the peaks over the Sea of Okhotsk and the west coast of the Bering
Sea. Several days later, the observed $SO_2$ over the west coast of the Bering Sea gradually spreads out and its vertical column
density gradually attenuates (not shown). In contrast, the two $SO_2$ peaks over the Sea of Okhotsk retained their compact
structures and relatively high vertical column density. From mid to late July, the two peaks over the Sea of Okhotsk eventually
comprise the main parts of the remaining $SO_2$ of the Raikoke eruption and developed into two compact $SO_2$ clouds. As the
simulated $SO_2$ did not reproduce the two peaks over the Sea of Okhotsk, however, the forward simulation lost its ability to
capture the observed $SO_2$ distribution during the first week of July.



**Figure 11.** TROPOMI observations and MPTRAC forward simulations of SO$_2$ transport. Top row (a-c): spatial distribution of SO$_2$ vertical column density (DU) from the TROPOMI orbit that starts at 23 June 2019, 01:05 UTC and ends at 23 June 2019, 02:47 UTC (a), and the corresponding distribution in forward simulations, which were initialized by TROPOMI observations (b) and by a constant injection rate (c), respectively. Middle and bottom rows show the same as the top row but for the orbits of 25 June 2019, 00:27-02:09 UTC and 28 June 2019, 01:12-02:53 UTC, respectively. The mean trajectories for injections between 7-8 km (red curve) and 11-15 km (black curves) are plotted on the maps on the left, and the mean locations at the corresponding time of each map are indicated by red and black dots.

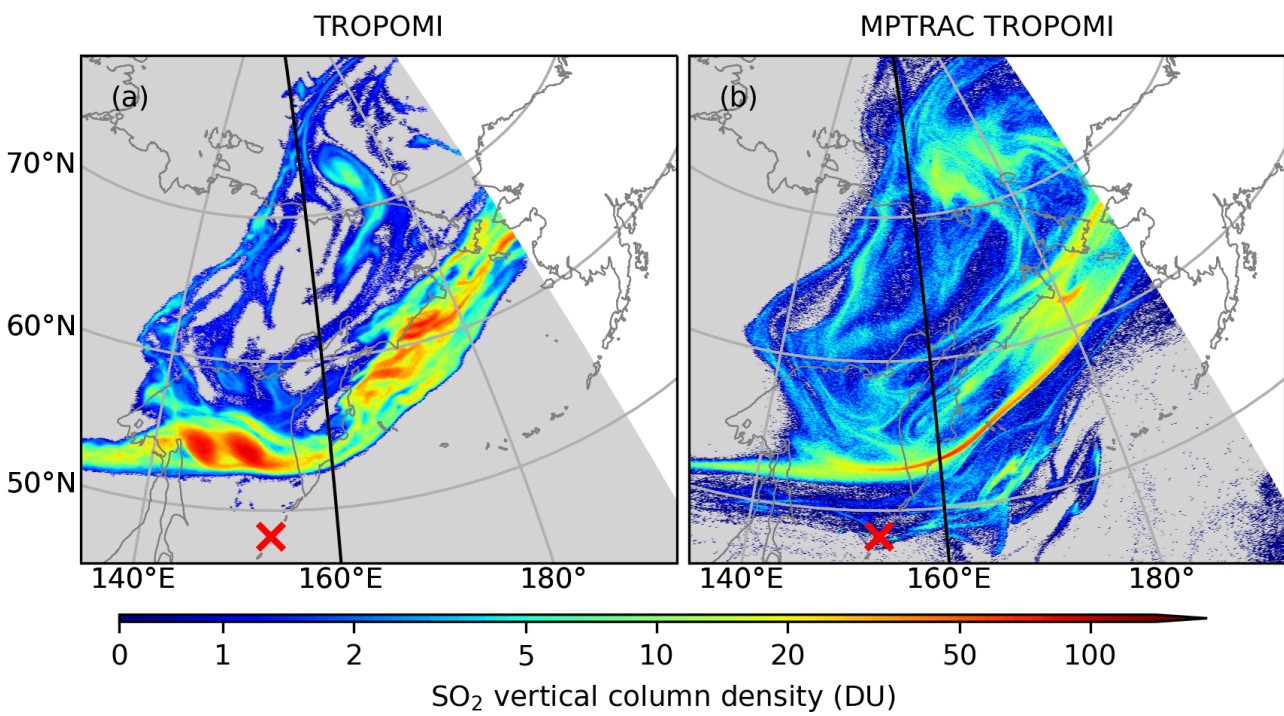

**Figure 12.** Spatial distribution of $SO_2$ vertical column density (DU) from the TROPOMI orbits that start at 1 July 2019, 00:15 UTC and end at 1 July 2019, 03:38 UTC (a), and the corresponding distribution in the forward simulation, which was initialized by TROPOMI observations (b). Note that data for the region in the east and west of $160°$E are from two neighboring orbits.





### 3.2.3 Assessment of forward simulations by means of the Critical Success Index

We performed analyses of the Critical Success Index (CSI) to evaluate the forward simulations at five different detection thresholds ranging from 0.3 to 50.0 DU (Figs. 13 and 14). Figure 13 shows the CSI, POD, and FAR time series for the forward simulation initialized with the TROPOMI observations. The reference for calculating the CSI, POD, and FAR are also the

TROPOMI observations to keep consistency of data between observations and simulations. The smallest threshold considered here represents the lower detection limit of the TROPOMI observations, which means that a threshold of 0.3 DU includes all available TROPOMI observations of the Raikoke event in the Northern Hemisphere. For the detection threshold of 0.3 DU the forward simulation produced a very high POD, being around 80% or larger. Such a high POD suggests that the overall spatial extent of the $SO_2$ distribution is well reproduced by the forward simulation. However, the FAR shows a significantly

increasing trend towards the end of the simulation, which suggests that the forward simulation transported some $SO_2$ outside of the observed $SO_2$ clouds. Due to the increasing trend of the FAR, the CSI values peak at the beginning of the simulation with a maximum value of 77% and gradually decrease to ~20% after 10 days. Compared with CSI analyses in previous studies on other volcanic eruption events (Heng et al., 2016; Hoffmann et al., 2016), in which the CSI decreased to below 10% after 10 days of forward simulations, our study for the Raikoke eruption shows improved performance of the model, meteorological

input data, and satellite observations.

When increasing the detection threshold, however, the model performance decreases. For instance, the POD shows a clear decreasing trend when the detection threshold is increased from 0.3 to 50 DU (Fig. 13a). The differences of the FAR between the different detection thresholds are smaller (Fig. 13b). This result suggests that although the forward simulation well reproduced the overall spatial extent of the $SO_2$ clouds, it has less ability to reproduce the internal structure and the location of the maxima

of the $SO_2$ clouds. For example, the POD at all $SO_2$ thresholds except for the 0.3 DU level decreased notably during the first week of July (Fig. 13a) agreeing with our earlier findings that the forward simulation has lost the ability to capture structures of the $SO_2$ clouds at this time.

For comparison, we also assessed the performance of the forward simulation initialized by the AIRS nighttime observations with reference to the AIRS nighttime observations (Fig. 14). As the AIRS observations have a higher background level (about

5 DU), the lowest detection threshold of 0.3 DU to assess the CSI is not very meaningful in this case. The trend and magnitude of the POD from AIRS observations are very similar to the simulation with TROPOMI observations, but the FAR has lower values of $20 - 40\%$. During the first 10 days of the simulation, the CSI values for a detection threshold of 5.0 DU are between 40-80%, which is about 1.5 times higher than for the simulation with TROPOMI observations.

To make a comparison between forward simulations with different initializations, we used the TROPOMI observations as

a common reference and the detection threshold was set to 5.0 DU. Figure 15 shows the POD, FAR, and CSI time series of forward simulations initialized with TROPOMI observations, AIRS nighttime observations, and a constant injection rate. In comparison to the TROPOMI observations, the POD, FAR, and CSI are very consistent between forward simulations initialized with TROPOMI and AIRS nighttime observations. Compared with the simulation with a constant injection rate, the POD is constantly higher for simulations initialized by observations. However, the simulations initialized with the satellite observations





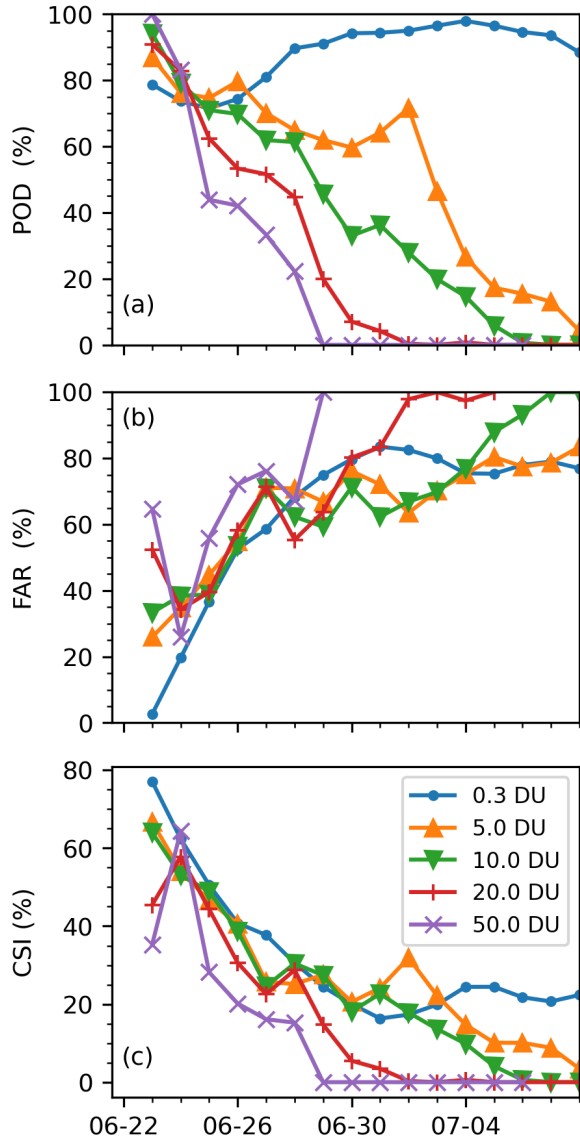

**Figure 13.** Time series of the probability of detection (POD; a), false alarm rate (FAR; b), and critical success index (CSI; a) for a forward simulation initialized by the reconstructed injection parameter based on TROPOMI data. The TROPOMI observations are used as reference observations.

suffer from higher FAR between 29 June and 4 July. In summary, the overall trends of POD, FAR, and CSI are generally similar between forward simulations with different initialization settings. This result indicates that the quality of the forward simulations is less affected by the injection parameters as estimated by the backward trajectory method, probably due to the fact that the major SO₂ injection occurred during a small time window at the beginning of the eruption in this case.

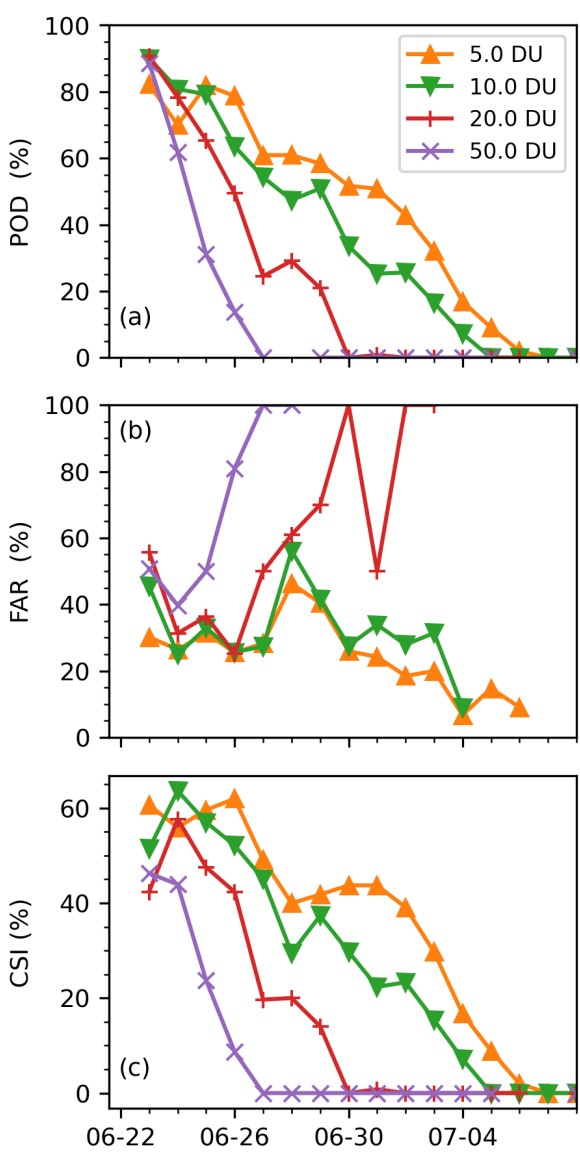

**Figure 14.** Same as Fig. 13, but for AIRS nighttime data. The AIRS observations are used as reference observations here.

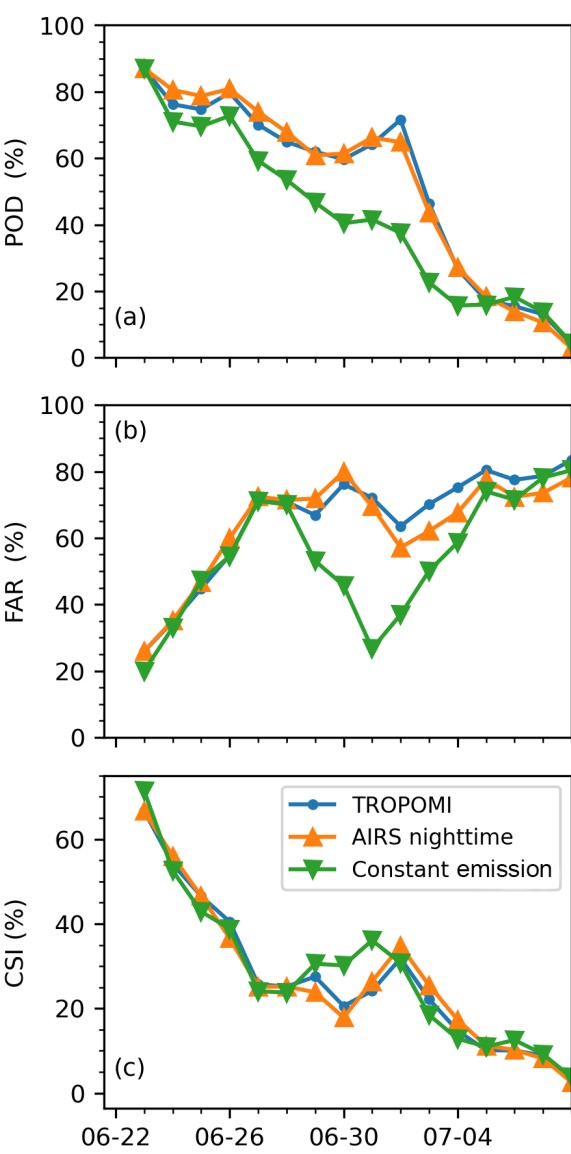

**Figure 15.** Time series of POD (a), FAR (b), and CSI (c) for forward simulations initialized by a constant injection rate and injection parameters reconstructed using TROPOMI and AIRS nighttime data, respectively. The reference observations are TROPOMI observations. Color coding indicates the column density threshold used to detect events (see plot key).





### 3.2.4 Simulation of compact SO$_2$ clouds from late July to early August

From early July, the injected SO$_2$ from the Raikoke eruption has gradually faded away and two compact SO$_2$ clouds became the major parts of the remaining SO$_2$. Figure 16 shows the location and distribution of the SO$_2$ clouds from 8 July to 9 August. During this period, the SO$_2$ is mainly concentrated in two compact clouds with a size of the magnitude of several hundred kilometers. On 8 July, the two compact SO$_2$ clouds are located close to each other, but eventually one of them moved toward the Asian continent (AC) and the other one moved toward North America (NA). In the following we refer to the SO$_2$ cloud that moved toward the Asian continent as the AC SO$_2$ cloud and the one toward North America as the NA SO$_2$ cloud.

The AC SO$_2$ cloud first moved westward toward the Raikoke volcano and moved over the volcano on 15 July. After that, the AC SO$_2$ cloud moved eastward and from 24 July it moved southwestward. After it reached 30°N on 29 July, it stayed at this latitude and moved westward. During the whole period between late July and early August, the AC SO$_2$ cloud remained confined in a compact structure. The AC SO$_2$ cloud with a confined structure remains detectable in satellite observations until late August and September 2019 (Chouza et al., 2020; Gorkavyi et al., 2021).

The unique structure of the AC SO$_2$ cloud motivated us to test the ability of the Lagrangian model in simulating the transport and dispersion of the SO$_2$ cloud, especially the parameterization of the dispersion processes. In each orbit where TROPOMI observed the AC SO$_2$ cloud, the SO$_2$ detections were re-sampled to a number of 10,000 air parcels with equal mass to represent the SO$_2$ cloud. The number of air parcels was scaled proportional to the SO$_2$ vertical column density. The altitude for all the observations is set to the altitude observed by the Cloud-Aerosol Lidar with Orthogonal Polarization (CALIOP) instrument (Gorkavyi et al., 2021). Note that CALIOP measures aerosol particles, which may introduce uncertainties regarding the altitude of gas phase SO$_2$. To reduce uncertainties associated with the vertical spread of the SO$_2$ in the simulations, the altitude of the air parcels corresponding to the same TROPOMI orbit was set to a constant value, i. e., the SO$_2$ was restricted to occur at the same altitude and no vertical spread was introduced during re-sampling. We used the re-sampled air parcels for TROPOMI observations during 17 July 2019 (Fig. 16) to initialize the forward simulation. Medians of the locations (longitudes and latitudes) of the air parcels are used to represent the location of the SO$_2$ cloud. The Median Absolute Deviation (MAD) is used to measure the degree of dispersion of the SO$_2$ cloud.

Qualitatively, we compared the simulated AC SO$_2$ cloud with observations at 1, 3, and 5 days after the initial release of the air parcels (Fig. 17). Dispersion in the simulations is in default settings, but different vertical motion schemes driven by vertical velocity (kinematic) and potential temperature (isentropic) are compared. As already shown in Fig. 16, the spatial extent of the AC SO$_2$ cloud is restricted in a limited bubble-like area during these times. One day after the start of the forward simulations, the model still well captures the spatial distribution of the AC SO$_2$ cloud (Fig. 17a and 17b). Despite the mean location being still captured by the model, the simulated SO$_2$ cloud is already too dispersive after 3 days. Although dispersion in both simulations is too strong, the simulation with constant isentropic vertical motions shows relatively weaker dispersion. In addition to dispersion, the simulated SO$_2$ cloud also shows some stretching effects along the west-southwest and east-northeast direction (Fig. 17g and 17h). Besides the horizontal location, however, both simulations driven by vertical velocity and constant potential temperature cannot correctly simulate the rising rate or the altitude of the SO$_2$ cloud (not shown). Although manual





**Figure 16.** Composite maps of the compact SO$_2$ clouds moving toward the Asian continent (a) and North America (b), respectively. The time of observation for each patch is indicated as text in the form of MMDDHH (Month-Day-Hour) of the year 2019.

corrections to the altitude position of the SO$_2$ cloud have been performed in a previous study (Gorkavyi et al., 2021), in our study the vertical motion is freely driven by vertical velocity or remained confined to a constant potential temperature.

The simulation of the dispersion of the AC SO$_2$ cloud was further evaluated by analyzing the MAD in longitude and latitude. Evaluation of the dispersion in the vertical direction is not available as the vertical spread of SO$_2$ is not fully observed. We designed separate experiments to test the role of subgrid scale variance and the role of diffusivity. When testing the subgrid variance, the diffusivities were set to zero, and vice versa when testing the subgrid variance. Fig. 18 shows the mean trajectories and the degree of dispersion of the AC SO$_2$ cloud in observations and forward simulations using different parameterizations of





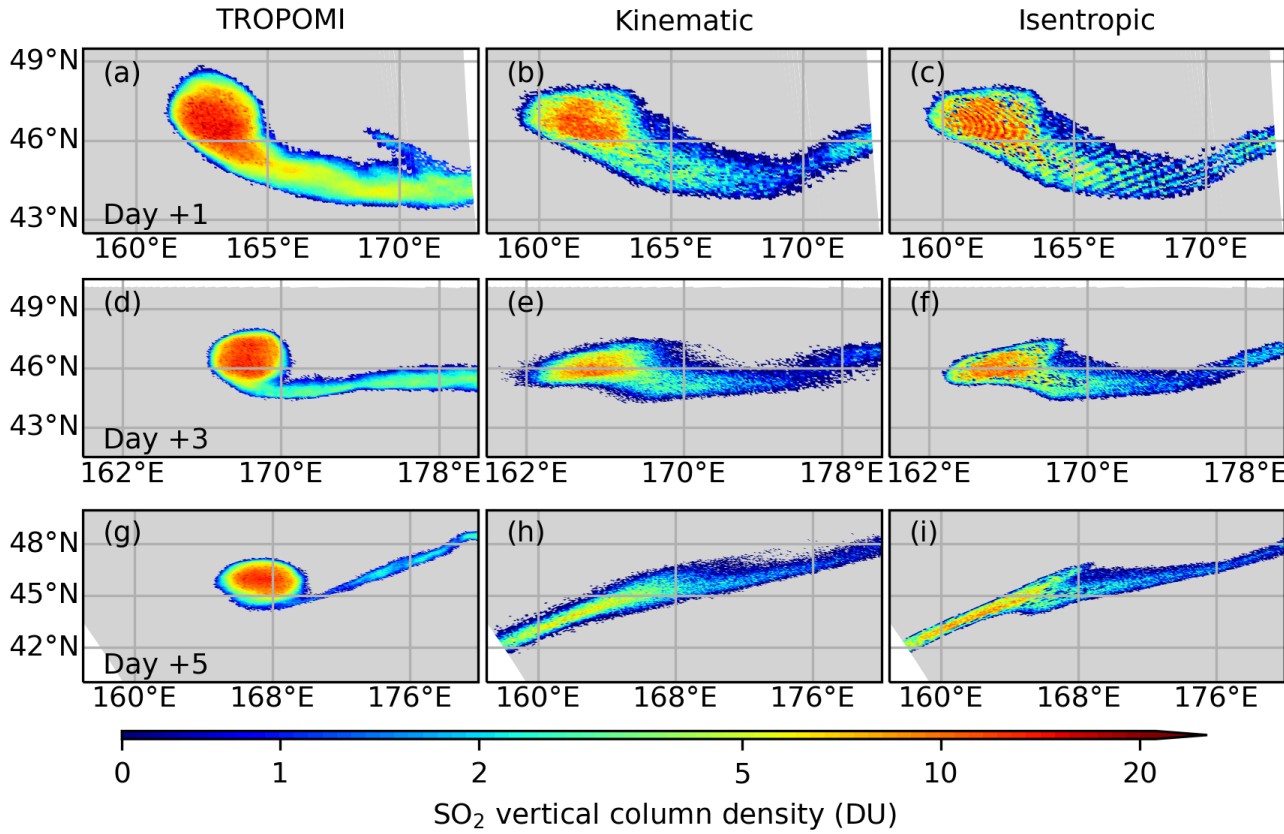

**Figure 17.** Forward simulation of the dispersion of the AC SO$_2$ cloud. Top row (a-c): spatial distribution of SO$_2$ vertical column density from TROPOMI at 18 July 2019, 02:33 UTC (a), and the corresponding distribution in forward simulations in which vertical motion was driven by vertical velocity (b) or by constant potential temperature (c), respectively. Middle and bottom rows are the same as the top row but for 20 July 2019, 01:55 UTC and 22 July 2019, 01:18 UTC, respectively.

the subgrid variance (the parameterizations on the horizontal and vertical directions are changed simultaneously). In general, all the forward simulations reproduce the trajectory of the observed AC SO$_2$ cloud (Fig. 18a). The MAD of the AC SO$_2$ in the TROPOMI observations is ∼1 degree in the longitude dimension and ∼0.6 degree in the latitude dimension. A common problem in the simulations is that the simulated dispersion is much stronger than the observed dispersion, even when the dispersion parameters were set to zero and just the initial horizontal spread of the cloud was taken into account. In the longitude

dimension, the MAD in simulations is roughly one order of magnitude larger than the MAD in observations after 5 days of simulation time. In contrast, in the latitude dimension the MAD in the simulation is similar to the MAD in the observation during the first ∼10 days of the simulation. After 10 days the MAD increased substantially in the simulations but not in the observations. Still, the differences of the MAD in latitude between observations and simulations are much smaller than the





MAD differences in longitude. This result again suggests a stretching effect due to horizontal wind shear along the longitude

direction beside the stochastic effects of dispersion.

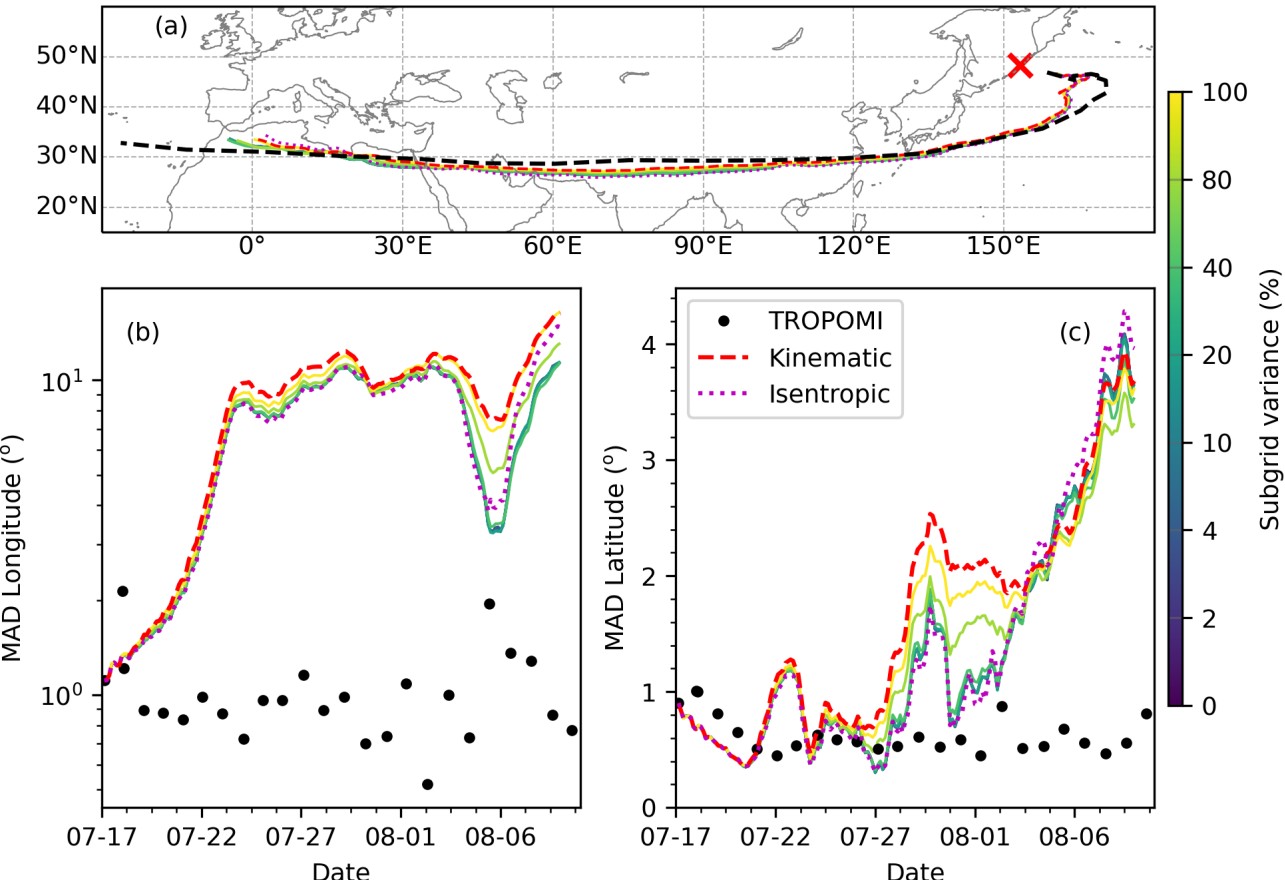

**Figure 18.** Trajectories (a) and degree of dispersion (b) in the longitude dimension and ( c) in the latitude dimension of the AC $SO_2$ cloud in simulations driven by vertical velocity, but with different subgrid scale diffusion settings (see color bar). Results from TROPOMI observations are shown in black. Results from the simulations (diffusion module is set to default in both simulations) with vertical motion driven by vertical velocity and potential temperature are shown in red and magenta, respectively.

We also tested the role of diffusivity ranging from $10^{-2}$ to $10^3$ times of the default diffusivity values (see Sect. 2.3 for reference). Similar to the results in Fig. 18, the simulated dispersion is too strong, but the simulation is more sensitive to diffusivity than subgrid scale variance (Fig. 19). When the diffusivity is on the order of $\geq 10$ times of the default diffusivity values, the simulation also does not capture the trajectory of the AC $SO_2$ cloud anymore.

We performed forward trajectory simulations with vertical motion driven by constant potential temperature to avoid potential vertical velocity fluctuations due to data assimilation. The degree of dispersion shows less sensitivity to the dispersion parameters and simulated MAD for longitude and latitude are very close to the results from the default setting (Fig. 18). As the

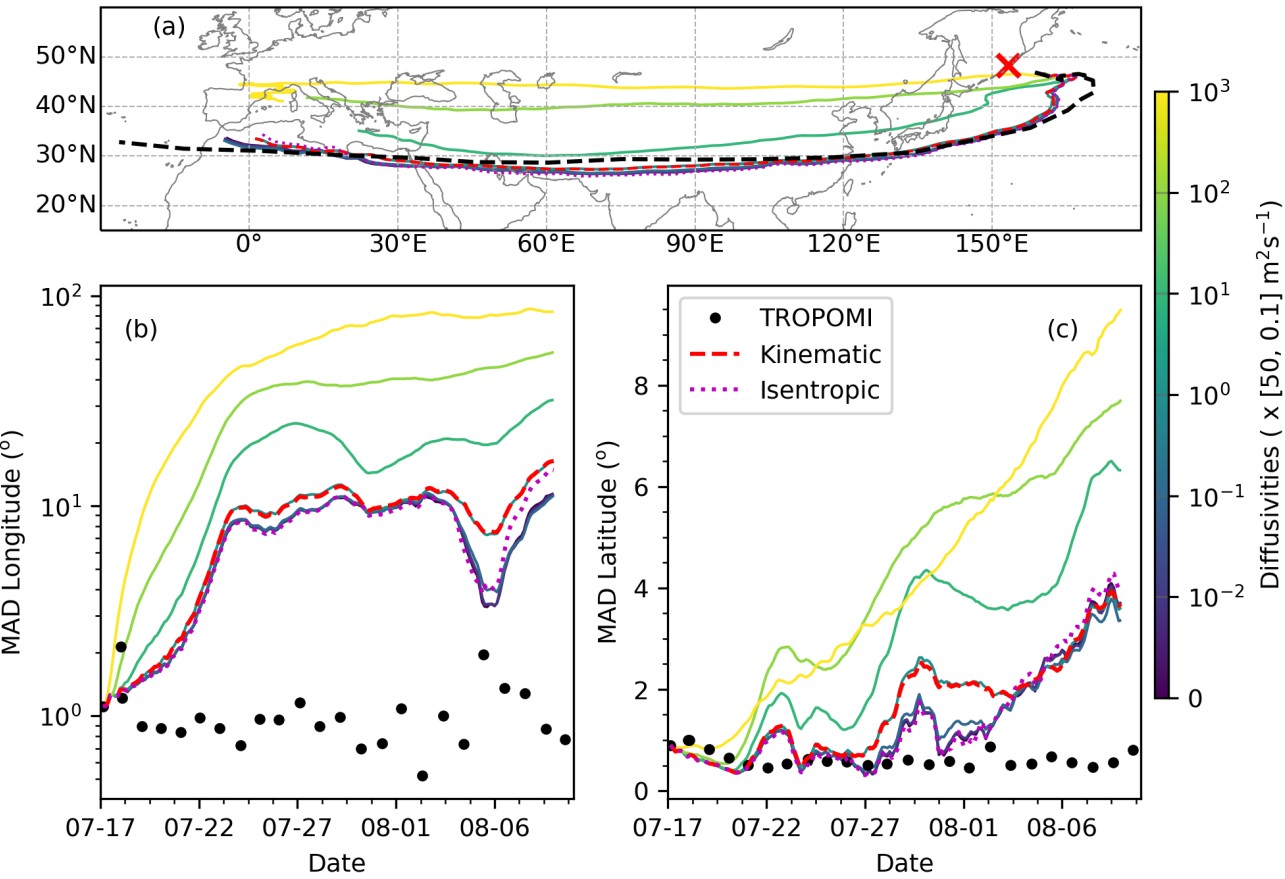

**Figure 19.** Same as Fig. 18, but for simulations driven by different diffusivity settings (see color bar).

vertical position is adjusted at every time step to retain a constant potential temperature, the dispersion in the vertical direction is suppressed in the isentropic mode. Further, as the AC SO$_2$ cloud is located in the stratosphere, the turbulent diffusion is

driven by vertical diffusivity, which is also switched off in the isentropic mode. Therefore, dispersion in the isentropic trajectory simulation is less sensitive to the choice of dispersion parameters and is weaker than the dispersion in kinematic mode. Taken together, our tests on the dispersion parameters suggest that the reason why the simulated SO$_2$ cloud generally is too dispersive is not only due to too strong diffusion in the model, but the stretching effect associated with horizontal wind shear also seems to play an important role.



## 4 Discussion

The injection parameters of a volcanic eruption, i. e., the time, altitude, and rate of $SO_2$ injections, have fundamental impacts on transport and dispersion simulations of volcanic ash and trace gases. Our study used two independent $SO_2$ satellite data products (from AIRS and TROPOMI, respectively) and a backward trajectory method implemented with the MPTRAC Lagrangian transport model to estimate the injection parameters of the 2019 Raikoke eruption. The reconstructed injection parameters generally agree with each other when different satellite datasets and meteorological reanalyses were used, indicating the robustness of our approach.

Our reconstruction shows that the $SO_2$ from the Raikoke eruption was mainly injected between 4-16 km of altitude, which falls within the range of previous studies (Hedelt et al., 2019; Muser et al., 2020; de Leeuw et al., 2021; Horváth et al., 2021). Similar to de Leeuw et al. (2021), our reconstruction also shows enhanced $SO_2$ concentrations at 12-14 km of altitude. Besides the overall agreement with the injection parameters used in this previous study, our reconstruction differs in some aspects. First, we estimated a total $SO_2$ injection of 2.1±0.2 Tg, which is larger than earlier estimates of 1.5±0.2 Tg (Global Volcanism Program, 2019; Muser et al., 2020; de Leeuw et al., 2021). Compared with the VolRes team estimation, de Leeuw et al. (2021) also argued that 1.5 Tg would underestimate the $SO_2$ mass in their forward simulation with the NAME model and they suggested that either more $SO_2$ should be injected into the stratosphere (1.09 Tg) or a total of 2.0 Tg would be needed to match the TROPOMI observations. The stratospheric injection in our reconstruction is ∼0.85 Tg, which is lower than the mass used by de Leeuw et al. (2021), but is significantly larger than the 0.64 Tg injection into the stratosphere in the VolRes profile.

Secondly, our reconstruction shows continuous but weak $SO_2$ injections after the first major injection on 21 − 22 June 2019. The major eruption during 21 − 22 June injects 1.5 − 1.6 Tg of $SO_2$ and a remaining fraction of 0.5 − 0.6 Tg of $SO_2$ was injected during 23 − 30 June mainly into the troposphere. Hedelt et al. (2019) also suggested minor injections on and after 23 June. A direct validation of injections after 22 June is difficult, as the injection rates are rather low. We inspected VIS and IR images from Himawari 8 (not shown) and found that some volcanic plumes are visible over the Raikoke during 24-25 June, corresponding to the second plume in our reconstruction. However, due to high altitude clouds it is hard to validate or rule out the possibility of a third plume. From our forward simulations, the second and third plume are potentially overestimated. However, totally excluding either of these injections would cause other problems in the simulations, in particular in simulating the $SO_2$ total mass. A future study, based on more sophisticated inverse modeling techniques (Heng et al., 2016), might yield an improved injection reconstruction.

Forward simulations using our reconstructed injection parameters compare well with the TROPOMI $SO_2$ observations during the first two weeks after the eruption in terms of location and spatial extent. Similar to the simulation using NAME (de Leeuw et al., 2021), our simulation also shows limited skills in capturing the structures inside the $SO_2$ clouds at a later stage. de Leeuw et al. (2021) argued that the limited ability to capture the internal structure of the $SO_2$ clouds is because the diffusion in the model is too strong. Hence, we explored the influence of the diffusion parameterization on the simulation of the compact $SO_2$ cloud from late July to early August as in (Gorkavyi et al., 2021). Although the simulation skill was improved when reducing the strength of the simulated diffusion, the simulation was still too diffusive. As the simulations were still too


diffusive when diffusion was switched off or when isentropic vertical motion were enforced to avoid jumps in the vertical velocity due to data assimilation (Stohl et al., 2005), we conclude that the strong dispersion is due to the meteorological input data itself. In particular, the stretching of the simulated $SO_2$ cloud (Fig. 17) and much stronger dispersion in the longitude direction (Fig. 18 and 19) suggest that the spread of the simulated $SO_2$ cloud is likely caused by horizontal wind shear in the ERA5 data. We did a similar set of experiments using ERA-Interim data, leading to the same results and conclusions.

Our study estimated the overall $SO_2$ e-folding lifetime during the first two weeks after the Raikoke eruption to be $13 - 17$
days. This finding was consistent between using a simple exponential decay of the reconstructed $SO_2$ injections and simulating chemical loss of $SO_2$ due to reaction with hydroxyl. The $SO_2$ mass burden derived from both, the exponential decay experiment and the hydroxyl module of MPTRAC, matches well with the TROPOMI observations. Our estimation also agrees with earlier studies. de Leeuw et al. (2021) estimated that the e-folding lifetime after 27 June is 14-15 days. Based on Ozone Mapping and Profiler Suite (OMPS) observations, Gorkavyi et al. (2021) estimated that the e-folding lifetime during the first 20 days after
the eruption is 18.9 days, which is slightly larger than the e-folding lifetime in our study and in de Leeuw et al. (2021). We note that, however, the e-folding lifetime has a strong dependence on the altitude of the $SO_2$ layer (Fig. 10), emphasizing that correctly determining the vertical profile of the injection rates is essential to reproduce the observed $SO_2$ mass change.

## 5  Conclusions

Determining the injection parameters of volcanic eruptions, including the plume altitude, time, and injection rate, is essential
for accurately simulating the dispersion of volcanic trace gases and aerosols. We used the MPTRAC model as well as AIRS and TROPOMI satellite observations to estimate the injection parameters of the 2019 Raikoke eruption. The altitude and time of the $SO_2$ injection was estimated based on a backward trajectory method and the $SO_2$ observations from the AIRS and TROPOMI satellites. Then, we used an exponential decay model to calibrate total injected $SO_2$ mass with the $SO_2$ mass from TROPOMI observations. The lifetime of $SO_2$ was estimated to be 13-17 days. Our estimation of the $SO_2$ mass change in the exponential
experiments agrees well with the mass change in the forward simulation that is driven by chemical reactions. Both methods reproduced the mass change derived from TROPOMI observations. Therefore, our method is robust for estimating the whole set of injection parameters, i. e., the time, altitude, and injection rate, for $SO_2$ injections.

Our estimated total $SO_2$ mass for the 2019 Raikoke eruption is 2.1 ± 0.2 Tg, which is larger than the initial estimate of 1.5 ± 0.2 Tg from earlier studies. We consider our new estimation of a larger amount of $SO_2$ reasonable, as it better reproduces
the observed mass change in forward simulations. The reconstructions of injection parameters are very consistent between using the TROPOMI daytime and AIRS nighttime observations. Forward simulations driven by our reconstructed time- and height-resolved injection parameters compared with simulations driven by a simple constant injection rate, an approach that is common in global chemistry-climate simulations, show better performance of reproducing the observations, especially in terms of spatial extent and location. The findings from this study will help us to create a long term volcanic $SO_2$ injection inventory
from AIRS, which we hope might be useful to improve chemistry climate simulations considering the effects of volcanic $SO_2$ in future work.



*Code and data availability.* The MPTRAC model is available under the terms and conditions of the GNU General Public License, Version 3 via the repository at https://github.com/slcs-jsc/mptrac. The TROPOMI SO$_2$ product data was obtained from the Copernicus Open Data Hub at https://scihub.copernicus.eu. The AIRS SO$_2$ data product used in this study was derived from the AIRS Level-1B data obtained from

NASA at https://doi.org/10.5067/YZEXEVN4JGGJ and is publicly available at https://datapub.fz-juelich.de/slcs/airs/volcanoes/. The ERA5 and ERA-Interim reanalysis data were obtained from ECMWF's Meteorological Archival and Retrieval System (MARS).

*Author contributions.* ZC, SG, and LH jointly developed the concept of this study. ZC performed the simulations and analyzed the data and results. SG prepared the TROPOMI data. LH provided the AIRS SO$_2$ data product and extended the MPTRAC Lagrangian transport model for the application in this study. ZC wrote the manuscript with contributions from all co-authors.

*Competing interests.* The authors declare that no competing interests are present.

*Acknowledgements.* We acknowledge the Juelich Supercomputing Centre for providing computing time and storage resources on the supercomputer JUWELS. Zhongyin Cai was partly supported by the International Postdoctoral Exchange Fellowship Program 2019 (grant no. 20191038), the Applied Basic Research Foundation of Yunnan Province (grant no. 202001BB050066), and the China Postdoctoral Science Foundation (grant no. 2019M653505). This research has been supported by the Deutsche Forschungsgemeinschaft (grant no. DFG

HO5102/1-1).





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
