# Peer review of "Improved estimation of volcanic $SO_2$ injections from satellite retrievals and Lagrangian transport simulations: the 2019 Raikoke eruption"

_Atmospheric Chemistry and Physics, 2021_

## Referee Comment (RC1)

Comment on "Improved estimation of volcanic SO2 injections from satellite observations and Lagrangian transport simulations: the 2019 Raikoke eruption" by Cai et al.

General comments:
Cai et al. describe an improved method to derive realistic time- and altitude-resolved volcanic SO2 emission rates based on satellite observations of SO2 and Lagrangian backward simulations. This study is the first application of the upgraded MPTRAC model in inverse modeling of volcanic SO2 injections and transport. By considering varieties of SO2 observations, adjusting the SO2 mass and initialization method, and including an OH chemical module, the updated procedure provided a more sophisticated way to retrieve explosive volcanic SO2 emission and achieved more promising results than their earlier version. The calibrations and sensitive tests make this study also a good tool article for the use of the MPTRAC model. The method described is of great potential in refining volcanic SO2 emissions in climate models. This manuscript is clearly organized and well written, so I recommend this study be published after minor revision.

My specific comments are as follows:
**Page 1**
    L1 are important
    L10 The reconstructed SO2 injection…
**Page 3**
    L71 observed from the satellite
    L78 since the beginning of operations
    L89 compact SO2 clouds
**Page 4**
    L106 5 Dobson Unit (DU)
**Page 5**
    Figure 1 caption: middle latitudes or mid-latitudes
**Page 6**
    Figure 2: 24-hour
    Figure 2: Are there gaps between tracks of AIRS and TROPOMI on the demonstrated area? If so, it would be easier for readers to distinguish between track gaps and areas with little SO2 with the track gaps indicated.
    L142 and hereafter, subgrid-scale
**Page 9**
    Does the thick black solid line indicate the altitude of the tropopause?
**Page 13**
    Figure 7: Although the cumulative SO2 emissions (Fig.5) from TROPOMI and AIRS nighttime are similar, the magnitude of the differences in emission rates in Fig. 7(a) seems almost as large as the emission rate in Fig.3(a). Based on your current results, could you conclude the best practices (including a suite of satellite data and exterior winds) for the Raikoke case?

L255–256 short-term, long-term, longer-term

**Page 15**

L277 Would you please specify the source data of hydroxyl radical or how the profile of hydroxyl radical is set for the OH chemistry module in the model?

**Page 25**

Figure 15: I guess there is a mistake in the figure caption. Figure 15 shows the POD, FAR, and CSI time series of forward simulations initialized with TROPOMI observations, AIRS nighttime observations, and a constant injection rate when the detection threshold was set to 5.0 DU.

**Page 26**

L410–414: Did you fix the SO2 column in a certain altitude level constrained by the altitudes of aerosols provided by CALIOP? Please make the altitude range clear, so readers do not have to search for and read Gorkavyi et al. only for the value of altitudes.

**Page 27**

L429 a constant potential temperature level/a isentropic surface.

**Others**

Please double-check the use of en dashes and hyphens between ranges of number and dates.

---

## Referee Comment (RC2)

Review of manuscript "Improved estimation of volcanic $SO_2$ injections from satellite observations and Lagrangian transport simulations: the 2019 Raikoke eruption" by Cai et al.

**General comments:**

The study by Cai et al. combines satellite retrievals from TROPOMI and AIRS with the Lagrangian transport model MPTRAC to give a detailed estimate of the volcanic $SO_2$ injections for the 2019 Raikoke eruption. By applying an inverse modelling technique, the authors give a detailed estimate of the time- and altitude-resolved $SO_2$ emission for this eruption. This study investigated a range of conditions in the latest version of MPTRAC (e.g., using various $SO_2$ retrievals, varying diffusion parameters, scaling of the initial $SO_2$ mass, including a new OH chemistry module), which results in a detailed consideration of the estimated $SO_2$ injections.

Initialising forward trajectories with this new $SO_2$ estimates for Raikoke can reproduce the $SO_2$ mass variations and spatial dispersion of the volcanic cloud retrieved by TROPOMI during the first 10 days after the eruption. Finally, the impact of diffusion is investigated, showing that it is too strong and as a result the model cannot capture the internal structure of the simulated $SO_2$ cloud well.

This work is very relevant to the atmospheric modelling community. Detailed eruption source parameters for volcanic eruptions are known to be difficult to determine. This study is a useful addition to the existing literature demonstrating that inverse modelling techniques are very useful to create more detailed eruption source parameters to initialise model simulations.

The manuscript is rather long, but I cannot see how to make it much shorter other than merging some of the figures. It is well written, and the figures are of a good quality and the authors give clear interpretations of the data. I have some minor concerns about some of the methodology used in this study, as outlined in the specific comments below. However, the overall work presented in the manuscript is good. I therefore recommend minor revisions to address the points outlined below before publication.

**Specific comments:**

L11: Satellites do not directly observe $SO_2$, but (as explained in sections 2.1 and 2.2) uses various bands in the infrared/UV spectra to retrieve estimates for $SO_2$. Therefore, in general it is better to use the terms 'retrievals/retrieved', rather than 'observations/observed' when discussing the satellite products. Please check carefully throughout the manuscript.

L114: "The AIRS… upper levels." I think this sentence needs a reference that supports this statement.

L115: 'particularly carefully'. Please avoid non-scientific terms. It should be clarified how the authors used the daytime data.

L130: In this study the results from the 15 km retrieval are presented. However, based on Fig. 3, you could argue that the assumed $SO_2$ height for the 15 km retrieval product is too high in the atmosphere for this eruption and that the 7 km retrieval from TROPOMI could be considered equally realistic (especially for the second and third phase). Did the authors investigate the impact of using the 7 km retrieval on their results? The 15 km and 7 km

retrieval products for TROPOMI can result in different $SO_2$ column mass estimates, which would in turn could also influence your estimate of the total emitted mass. It would be very interesting to understand if you would still get the reported 2.1±0.2Tg estimate when applying the 7 km retrieval. I think some discussion on this potential source of uncertainty should be included in the paper.

L155: "… reaction with the hydroxyl radical (OH)." What data is used to provide the OH field for the MPTRAC simulations?

L179-190: The work presented here samples trajectories from each satellite footprint between 0-25 km and combine them to obtain a best estimate of the $SO_2$ emission. However, how certain is it that there is a single best solution to the problem? Is it possible that a different emission profile not found by this method could give an equally good comparison with the TROPOMI retrievals?

For example, assume two hypothetical layers (say at 2 km and 7 km) at a given TROPOMI pixel location that both can be retraced to the volcano. I am not sure I understand how the backward trajectory method is able to determine which layer contained the $SO_2$ mass (or is able to determine the ratio between the two layers)? If I understand correctly, based on the sensitivity of the TROPOMI satellite (fig. 1), the method seems to be biased towards the higher altitudes, as more trajectories are released from the higher layers (7 km in our example case). But in our example, it is also possible that the mass was emitted in the lower layer, as it would give the same column total mass in TROPOMI. Is the backward trajectory method able exclude either possibility?

L193: "fixed e-folding lifetime". How realistic is it to use a constant e-folding lifetime for the entire altitude range considered? As mentioned by the authors (L.296-302, Fig.10), the lifetime of $SO_2$ varies strongly between the troposphere and the stratosphere. How should we interpret this fixed e-folding lifetime? Is it an altitude-weighted average lifetime for the $SO_2$?

L199: "To achieve a total injection of 2.1 Tg." So far, no evidence is given why 2.1 Tg would be more realistic. I think the authors should point to the discussion presented in section 3.1.2 or give a short explanation here on why the results have been tuned to 2.1 Tg.

Fig.3: What does the black line in panel 3a represent?

L202: "…prominent second and third plume…" Looking at the video's provided in the study by de Leeuw et al, their dispersion simulations show that part of the first plume is dispersed back to the location of the Raikoke volcano around the same time this study identified the third plume (27-28 June). This is also shown by the red trajectory in figure 11g. How would the backward trajectory methodology deal which such an event and is it possible that this third peak in the shown analysis is partly a reoccurrence of the first plume at the Raikoke location. If back trajectories hit the source location multiple times, would it pick the first hit only, identifying a 'new' source, or would it consider the possibility of multiple overpasses over the volcano at earlier times?

Related to this point, the video for $SO_2$ dispersion provided in the study by de Leeuw et al. also shows that part of the $SO_2$ cloud (using the VolRes profile) initially moves into a North-West direction, followed by it moving back and across the Raikoke volcano location around the 25th of June. Based Fig.11 in the current paper, this part of the plume is not present in the

presented dispersion experiment (panel 11b), while it is visible on the TROPOMI retrieval (panel 11a). This suggests that it might be linked to emissions at low altitudes (<5km). I wonder if part of the 2nd plume in fig. 3 could be related to this residual of the low altitude first plume that moves across the Raikoke volcano during the initial few days after the eruption and is misinterpreted as an additional emission of $SO_2$ at higher altitudes (see also point L179)?

I think it could be a great addition to the paper if it is possible to initialise the MPTRAC model using the VolRes profile as input and compare it with the results presented in this study (e.g. figs. 9, 11, 15).

L203: How is the tropopause defined?

L248: "…it matters how many days of satellite observations are used…". How many days are used for the results presented in this study (e.g. figure 3)? I think this should be specified in the manuscript. I also think some additional explanation is missing that describes how multiple days of the TROPOMI satellite retrievals are combined to reconstruct the $SO_2$ injections. When using 12 days of retrievals, does this mean that all the earlier overpasses are still considered? Or does this mean that the back-trajectories are calculated for 12 days to reconstruct the signal for this specific overpass?

Figure 9: Would it be possible to extent the figure to longer timescales? Based on figures 3, 10 and 13-15 the data is available, so I wonder why it wasn't included here?

L270: Why did the authors choose to implement the constant injection rate to represent 1.5 Tg and not 2.1 Tg like the other simulations? Earlier it is established that 1.5 Tg does not give realistic values, so is it considered for the constant injection rate. When looking at figure 9, moving up the constant emission simulation by 0.6 Tg, apart from the initial peak, you get a much better comparison with TROPOMI and the results for this simulation also fall within the satellite uncertainty range after several days, like the other two simulations.

Also, it is not clear to me if the chemistry module is used for the constant injection rate simulation or whether it uses the 14-day exponential decay (similar to the orange line in figure 9). Based on how smooth the removal is in figure 9 for the purple line, I think the latter is true, but this needs to be clarified.

L277: Which chemical reactions are included in the OH chemistry module and what are the reaction rates? I think a description of the chemistry module or a citation explaining the module should be included.

L283: "… are consistent with the total $SO_2$ mass derived from TROPOMI estimations." If I interpret figure 8 correctly, it seems that the MPTRAC data in figure 9a are initialised using the TROPOMI retrievals during the first 10-12 days. Therefore, I wonder how independent the two datasets are for the period shown in the figure and whether the very good agreement is a direct consequence of the way how the MPTRAC simulations are initialised using the same retrievals to which it is now again compared against.

L300-305: The constant injection rate simulation assumes that the mass is emitted uniformly between 5 and 15 km altitude. Assuming a tropopause at 10 km, this means that approximately 50% of the mass is emitted into the troposphere, where the lifetime of $SO_2$ is

much shorter (as seen in figure 10). If you would use a constant injection rate that has a more realistic profile with more of the mass emitted into the lower stratosphere/upper troposphere, how much would this improve the comparison?

L327: "… mainly associated with transport of $SO_2$ in the lower troposphere (between 0 and 5km), which was not represented in both initializations". Why did the backward trajectory method not manage to track back the TROPOMI footprints to the volcano for this part of the plume? Based on figures 11a and b, most of the southern branch retrieved by TROPOMI is not present in the MPTRAC simulations. The fact that the method does not seem to capture this large area of the plume associated with the lower emission altitudes makes me doubt the robustness of the method (see point L179), especially for lower altitude eruptions? I think this potential limitation should be discussed in the discussion section.

L333: Would it be possible that the part not represented by the constant emission injection rate simulation in panel 11f is linked to emissions at lower altitudes that were also missing in panels 11b-c? I think it would be very useful to repeat the analysis for the VolRes profile (which includes the lower-level emissions) to see if this could explain part of the differences observed.

Fig. 13-14: I think these figures can be combined to one figure with 6 panels, rather than having two separate figures. Label of Fig.13: (CSI, a) -> (CSI, c)

Fig. 15: 'Color coding indicates the column density threshold…' This is incorrect, as the different colors show the different simulation initialisations at a constant detection threshold (5 DU).

L410-414: I think the authors should include the altitudes used in MPTRAC for this part of the analysis. Maybe in a table in the supplementary material could be a good option, as the current manuscript is pretty long already.

L460: I miss a short discussion of the limitations/uncertainties related to the MPTRAC model and the backward trajectory method. One potential impact not discussed is the impact of the lofting of the plume due to the co-existence of ash. In the paper by Muser et al. 2020, a lofting effect was identified for the Raikoke plume during the initial days after the eruption. I can't find any information that the MPTRAC model accounts for this effect and as a result the back-trajectories could be placed at the wrong altitudes in the reconstruction. This in turn could have an impact on the forward simulations on longer timescales.

L474: The study by de Leeuw et al. shows that the 2.0 Tg emission profile overestimates the $SO_2$ mass from the TROPOMI retrievals during the first few days after the eruption. Therefore, I think this statement would be more accurate when adding 'to match the TROPOMI retrievals on timescales > 1 week'

L475: Are the stratospheric amounts similar for all the simulations considered or is this the value for the most accurate simulation? If increasing the emission to 2.1 Tg for the constant emission case, how much would be emitted into the stratosphere? I think it would be a very useful addition if the authors could include an uncertainty range for the 0.85 Tg estimate using the full range of simulations they have performed.

L483: "From our forward simulations, the second and third plume are potentially overestimated." Would you be able to identify potential reasons for this overestimation? I think a short discussion what might have caused the overestimation should be included here.

L488 and L499: 2 weeks -> 10 days. In this study only the location and spatial extend during first 10 days are discussed.

L518: I think it would be useful to include the fraction of the mass emitted into the stratosphere (0.85 Tg) here? Especially for climate impact studies, it is mainly the long-term stratospheric part of the plume that is of particular interest, as most of the tropospheric signal will be removed after several days/week.

L519: Better than what? The study does not show how the presented mass estimates compare with mass estimates using the profiles from other studies.

**Technical corrections/suggestions:**

Order of the figure panels. Some figures have panel b above a (fig. 3) and others have panel b below panel a (fig. 7). Please use one consistent ordering of labelling the panels in the figures to avoid confusion.

L40.: Besides -> Beside

L41: parcels -> parcel

L71: observed from satellite -> retrieved by the satellite

L78: since begin of -> since the beginning of

L85: satellite observations -> satellite retrievals

L106: DU -> Dobson Unit (DU)

L178: As both, AIRS and TROPOMI -> As both AIRS and TROPOMI

L199: turned -> tuned

Fig.7: Differences of -> Differences in

Fig.7: TROPIMI -> TROPOMI

L255: short term -> short-term

L256: long term -> long-term

References: Please check all the references carefully, as some have the DOI included twice (e.g. Hoffman et al 2014). Also use abbreviations for all the journals consistently.

---

## Author Comment (AC1)

**Reply to review comments**

We thank the reviewers for the time and effort spent on the manuscript and for providing helpful comments. We considered all comments and hope that the revised draft properly addresses the open issues. Please find our point-by-point replies below (colored in blue).

**Reviewer #1**

General comments:

Cai et al. describe an improved method to derive realistic time- and altitude-resolved volcanic $SO_2$ emission rates based on satellite observations of $SO_2$ and Lagrangian backward simulations. This study is the first application of the upgraded MPTRAC model in inverse modeling of volcanic $SO_2$ injections and transport. By considering varieties of $SO_2$ observations, adjusting the $SO_2$ mass and initialization method, and including an OH chemical module, the updated procedure provided a more sophisticated way to retrieve explosive volcanic $SO_2$ emission and achieved more promising results than their earlier version. The calibrations and sensitive tests make this study also a good tool article for the use of the MPTRAC model. The method described is of great potential in refining volcanic $SO_2$ emissions in climate models. This manuscript is clearly organized and well written, so I recommend this study be published after minor revision.

We thank the reviewer for the overall positive comments and have revised the manuscript according to the comments and suggestions provided by the reviewer. Please see a detailed reply to each comment below on the revisions in the revised manuscript.

My specific comments are as follows:

L1 are important

Corrected.

L10 The reconstructed $SO_2$ injection...

Added "$SO_2$".

L71 observed from the satellite

Changed into "retrieved by the satellite".

L78 since the beginning of operations

Corrected.

L89 compact SO$_2$ clouds

Corrected.

L106 5 Dobson Unit (DU)

Fixed.

Figure 1 caption: middle latitudes or mid-latitudes

Changed into "mid-latitudes".

Figure 2: 24-hour

Corrected.

Figure 2: Are there gaps between tracks of AIRS and TROPOMI on the demonstrated area? If so, it would be easier for readers to distinguish between track gaps and areas with little SO$_2$ with the track gaps indicated.

For TROPOMI, there are no gaps between tracks on the demonstrated area. For AIRS, only minor gaps exist between tracks in the region of 40-50N. However, the SO$_2$ cloud was not located in the gaps between tracks on the selected day. Therefore, the tracks were not shown in the figure.

L142 and hereafter, subgrid-scale

Changed throughout the manuscript.

Does the thick black solid line indicate the altitude of the tropopause?

Yes, it is the altitude of the tropopause calculated based on the WMO temperature lapse-rate definition. This information has been added in the revised figure caption.

Figure 7: Although the cumulative SO$_2$ emissions (Fig.5) from TROPOMI and AIRS nighttime are similar, the magnitude of the differences in emission rates in Fig. 7(a) seems almost as large as the emission rate in Fig. 3(a). Based on your current results, could you conclude the best practices (including a suite of satellite data and exterior winds) for the Raikoke case?

We cannot give a recommendation on the best choice of satellite and meteorological data to be used for estimating the SO$_2$ injections. However, our results indicate that the cumulative masses are robust, irrespective of the selected satellite and wind data set. The

significant differences in Fig. 5 mainly arise from differences in the timing of the reconstructed injections. Our recommendation would be to conduct tests with different data sets to assess their impact on the results. Such tests have been done in the current paper as well as previously by Hoffmann et al. (2016) and Kristiansen et al. (2012)..

L255–256 short-term, long-term, longer-term

Changed throughout the manuscript.

L277 Would you please specify the source data of hydroxyl radical or how the profile of hydroxyl radical is set for the OH chemistry module in the model?

To clarify, we added: "Monthly mean zonal mean OH concentrations are obtained from the study of Pommrich et al. (2014)."

Figure 15: I guess there is a mistake in the figure caption. Figure 15 shows the POD, FAR, and CSI time series of forward simulations initialized with TROPOMI observations, AIRS nighttime observations, and a constant injection rate when the detection threshold was set to 5.0 DU.

Yes, there was a mistake in the figure caption. "Color coding indicates the column density threshold used to detect events (see plot key)." has been removed, and "The column density threshold used to detect events is 5.0 DU." has been added.

L410–414: Did you fix the $SO_2$ column in a certain altitude level constrained by the altitudes of aerosols provided by CALIOP? Please make the altitude range clear, so readers do not have to search for and read Gorkavyi et al. only for the value of altitudes.

Yes, the $SO_2$ column are fixed to a certain altitude level constrained by the altitudes of aerosols provided by CALIOP. In the revised manuscript, we have added information on the value of altitudes during the study period: "During 17 July to 21 July 2019, the aerosol altitude is $\sim$18 km, and it rises to $\sim$20 km during 24-27 July, after which it gradually rises to 24 km around 14 August 2019 (Gorkavyi et al., 2021)."

L429 a constant potential temperature level/a isentropic surface.

Corrected.

**Others**

Please double-check the use of en dashes and hyphens between ranges of number and dates.

It is fixed now.

**Reviewer #2**

General comments:

The study by Cai et al. combines satellite retrievals from TROPOMI and AIRS with the Lagrangian transport model MPTRAC to give a detailed estimate of the volcanic $SO_2$ injections for the 2019 Raikoke eruption. By applying an inverse modelling technique, the authors give a detailed estimate of the time- and altitude-resolved $SO_2$ emission for this eruption. This study investigated a range of conditions in the latest version of MPTRAC (e.g., using various $SO_2$ retrievals, varying diffusion parameters, scaling of the initial $SO_2$ mass, including a new OH chemistry module), which results in a detailed consideration of the estimated $SO_2$ injections.

Initialising forward trajectories with this new $SO_2$ estimates for Raikoke can reproduce the $SO_2$ mass variations and spatial dispersion of the volcanic cloud retrieved by TROPOMI during the first 10 days after the eruption. Finally, the impact of diffusion is investigated, showing that it is too strong and as a result the model cannot capture the internal structure of the simulated $SO_2$ cloud well.

This work is very relevant to the atmospheric modelling community. Detailed eruption source parameters for volcanic eruptions are known to be difficult to determine. This study is a useful addition to the existing literature demonstrating that inverse modelling techniques are very useful to create more detailed eruption source parameters to initialise model simulations.

The manuscript is rather long, but I cannot see how to make it much shorter other than merging some of the figures. It is well written, and the figures are of a good quality and the authors give clear interpretations of the data. I have some minor concerns about some of the methodology used in this study, as outlined in the specific comments below. However, the overall work presented in the manuscript is good. I therefore recommend minor revisions to address the points outlined below before publication.

We thank the reviewer for the constructive comments and the detailed suggestions on the manuscript. We have studied each of the comments and suggestions and revised the manuscript accordingly. The discussion on the limitations and uncertainties associated with the current study is strengthened in the revised manuscript. Please see below the replies to each comment/suggestion and the corresponding revisions in the revised manuscript.

Specific comments:

L11: Satellites do not directly observe $SO_2$ , but (as explained in sections 2.1 and 2.2) uses various bands in the infrared/UV spectra to retrieve estimates for $SO_2$. Therefore, in general it is better to use the terms 'retrievals/retrieved', rather than 'observations/observed' when discussing the satellite products. Please check carefully throughout the manuscript.

Following the suggestion, we have checked the use of "observations/observed" throughout

the manuscript and have changed them into "retrievals/retrieved".

L114: "The AIRS... upper levels." I think this sentence needs a reference that supports this statement.

We found that Prata et al. (2010) pointed out that the effect of scattered solar radiation on the $7.3\,\mu$m waveband of $SO_2$ is negligible. Therefore, we removed this sentence with our hypothesis from the paper.

L115: 'particularly carefully'. Please avoid non-scientific terms. It should be clarified how the authors used the daytime data.

We would like to stress that AIRS day- and nighttime data should not be mixed up in the analysis. We shortened the sentence to "Therefore, the AIRS nighttime and daytime data are considered separately in this study.", to clarify.

L130: In this study the results from the 15 km retrieval are presented. However, based on Fig. 3, you could argue that the assumed $SO_2$ height for the 15 km retrieval product is too high in the atmosphere for this eruption and that the 7 km retrieval from TROPOMI could be considered equally realistic (especially for the second and third phase). Did the authors investigate the impact of using the 7 km retrieval on their results? The 15 km and 7 km retrieval products for TROPOMI can result in different $SO_2$ column mass estimates, which would in turn could also influence your estimate of the total emitted mass. It would be very interesting to understand if you would still get the reported 2.1±0.2Tg estimate when applying the 7 km retrieval. I think some discussion on this potential source of uncertainty should be included in the paper.

We have previously looked at the different TROPOMI $SO_2$ products that assume the $SO_2$ layer is at either 1, 7, or 15 km above sea level. Although the different retrievals assumed different $SO_2$ layer height, the vertical column density product itself does not contain altitude information. The main difference between different products is the absolute value of the vertical column density. Therefore, it does not influence the reconstruction of the relative injection rate. But when calibrating the absolute $SO_2$ mass (section 3.1.2), the different product could have an impact due to difference in the absolute $SO_2$ column density. Comparison of the total $SO_2$ mass between the 7 km and 15 km retrieval products for TROPOMI show that the mass is identical during the first week of the eruption and then the 7 km product gets an overall higher mass estimate. After the first week, the mass derived from the 7 km product is consistently higher than 15 km product by $\sim$10 percent. Therefore, using the 7 km product would get an higher estimate of the total $SO_2$ mass but it is still within our reported uncertainty range. The following text has been added to the revised text: "Besides the above limitations, the current reconstruction and in turn the forward simulations may be also influenced by the selection of the TROPOMI products, i.e. the altitude of assumed $SO_2$ layer during retrieval, and by the lofting of the plume due to the co-existence of ash. TROPOMI $SO_2$ products are available for different scenarios that assume the $SO_2$ is at either 1, 7, or 15 km above sea level. The main difference between different products is the absolute value of the vertical column density, and it has minor

influence on the reconstruction of the relative injection rate. However, the different $SO_2$ can result in different mass estimate. Comparison of the total $SO_2$ mass between the 7 km and 15 km retrieval products for TROPOMI show that the mass is identical during the first week of the eruption. After the first week, the mass derived from the 7 km product is consistently higher than the 15 km product by $\sim$10 percent. Therefore, using the 7 km product would get an higher estimate of the total $SO_2$ mass which is at the upper limit of the estimate reported in this study."

L155: "... reaction with the hydroxyl radical (OH)." What data is used to provide the OH field for the MPTRAC simulations?

To clarify, we added: "Monthly mean zonal mean OH concentrations are obtained from the study of Pommrich et al. (2014)."

L179-190: The work presented here samples trajectories from each satellite footprint between 0-25 km and combine them to obtain a best estimate of the $SO_2$ emission. However, how certain is it that there is a single best solution to the problem? Is it possible that a different emission profile not found by this method could give an equally good comparison with the TROPOMI retrievals? For example, assume two hypothetical layers (say at 2 km and 7 km) at a given TROPOMI pixel location that both can be retraced to the volcano. I am not sure I understand how the backward trajectory method is able to determine which layer contained the $SO_2$ mass (or is able to determine the ratio between the two layers)? If I understand correctly, based on the sensitivity of the TROPOMI satellite (fig. 1), the method seems to be biased towards the higher altitudes, as more trajectories are released from the higher layers (7 km in our example case). But in our example, it is also possible that the mass was emitted in the lower layer, as it would give the same column total mass in TROPOMI. Is the backward trajectory method able exclude either possibility?

Yes, it is possible that for a given TROPOMI pixel location there may be multiple solutions in the backward trajectory method, especially if the wind does not change with height and over time. Taking your example and assuming that the wind at 2 and 7 km is the same, but different in-between. In the first step at all altitudes the same number of air parcels is initialized. Applying the TROPOMI kernel function redistributes the air parcels such, that about twice as much air parcels would start at 7 km compared to 2 km. MPTRAC would trace the air parcels at 2 and 7 km back to the volcano. So, in this case there is the hypothetical chance to falsely attribute a large $SO_2$ mass at 2 km to 7 km, if only the layer at 2 km contains the volcanic plume. Fortunately the wind changes with height and time. In the study, this uncertainty has been reduced through two ways. First, TROPOMI retrievals over many days, in the final reconstruction 12 days, were used and second, an increased number of particles, 5 million air parcels, were released in the backward run. To make this clearer we added the aforementioned discussion to section 2.4.

L193: "fixed e-folding lifetime". How realistic is it to use a constant e-folding lifetime for the entire altitude range considered? As mentioned by the authors (L.296-302, Fig.10), the lifetime of $SO_2$ varies strongly between the troposphere and the stratosphere. How should

we interpret this fixed e-folding lifetime? Is it an altitude-weighted average lifetime for the $SO_2$ ?

The e-folding lifetime assumed here is an "altitude-weighted average lifetime" representing the overall removal of the $SO_2$ in the atmosphere. The value was empirically determined here and more detailed information on it should refer to later sections of the paper (including also Fig. 10). We have revised the text in the manuscript to clarify.

L199: "To achieve a total injection of 2.1 Tg." So far, no evidence is given why 2.1 Tg would be more realistic. I think the authors should point to the discussion presented in section 3.1.2 or give a short explanation here on why the results have been tuned to 2.1 Tg. Fig.3: What does the black line in panel 3a represent?

Following the reviewer's suggestion, we have added a forward reference to Section 3.1.2 in the revised text. The black line in panel 3a represents the ERA5 temperature lapse-rate tropopause over Raikoke volcano. We have added this information to the revised figure caption.

L202: "...prominent second and third plume..." Looking at the video's provided in the study by de Leeuw et al, their dispersion simulations show that part of the first plume is dispersed back to the location of the Raikoke volcano around the same time this study identified the third plume (27-28 June). This is also shown by the red trajectory in figure 11g. How would the backward trajectory methodology deal which such an event and is it possible that this third peak in the shown analysis is partly a reoccurrence of the first plume at the Raikoke location. If back trajectories hit the source location multiple times, would it pick the first hit only, identifying a 'new' source, or would it consider the possibility of multiple overpasses over the volcano at earlier times? Related to this point, the video for $SO_2$ dispersion provided in the study by de Leeuw et al. also shows that part of the $SO_2$ cloud (using the VolRes profile) initially moves into a North-West direction, followed by it moving back and across the Raikoke volcano location around the 25 th of June. Based Fig.11 in the current paper, this part of the plume is not present in the presented dispersion experiment (panel 11b), while it is visible on the TROPOMI retrieval (panel 11a). This suggests that it might be linked to emissions at low altitudes (¡5km). I wonder if part of the 2 nd plume in fig. 3 could be related to this residual of the low altitude first plume that moves across the Raikoke volcano during the initial few days after the eruption and is misinterpreted as an additional emission of $SO_2$ at higher altitudes (see also point L179)? I think it could be a great addition to the paper if it is possible to initialise the MPTRAC model using the VolRes profile as input and compare it with the results presented in this study (e.g. figs. 9, 11, 15).

In the current setting, the method would only pick the first hit to identify a new source and the second overpass will be not counted anymore. Regarding this question, we had developed a module to constrain the method only picks the hit that happens within a certain time range. However, after tests with some simulations we choose to present the current result that using a more straightforward method. Based on satellite images from

TROPOMI, Himawari 8, as well as the SACS web site, it looks like the 2nd plume is "real" and visible in the different satellite images. However, the $SO_2$ concentrations are very low (¡ 2-3 DU), i.e., would not be visible to AIRS and might be located at low altitudes (¡ 5 km), as pointed out by the reviewer. We also have less confidence on the third plume. For the consideration of mass, keep the third plume would yield a better agreement with the TROPOMI retrievals. In addition, as the second and third plume are located in the troposphere and quickly removed, and would not have any large impact on climate, they are not so important in the end. In the earlier phase of this project, we also tried to use the VolRes profile. However, as de Leeuw et al. (2021) already done extensive tests using the VolRes profile and our results using the VolRes are similar with de Leeuw et al. (2021), we did not show the results using VolRes profile, but using rather a more simple "guess" of a constant injection rate. In the end, our results are not so different from VolRes profile (Figure 4), despite the approach and data are rather different. Due to the similarity of the vertical profile, comparison between forward simulations based on our reconstruction and VolRes profile also contains less information. In the revised text, we have strengthened the discussion on the uncertainty regarding the second and third plume and the forward simulations using the VolRes profile de Leeuw et al. (2021).

L203: How is the tropopause defined?

We applied the thermal lapse-rate tropopause definition of WMO (1957). This information has been added to the text. Additional information on the ERA5 tropopause data set applied here has been added to the code and data availability section of the paper. The sentence reads now: "The first plume crossed the temperature lapse-rate tropopause (WMO, 1957) and injected $SO_2$ ..."

L248: "...it matters how many days of satellite observations are used...". How many days are used for the results presented in this study (e.g. figure 3)? I think this should be specified in the manuscript. I also think some additional explanation is missing that describes how multiple days of the TROPOMI satellite retrievals are combined to reconstruct the $SO_2$ injections. When using 12 days of retrievals, does this mean that all the earlier overpasses are still considered? Or does this mean that the back-trajectories are calculated for 12 days to reconstruct the signal for this specific overpass?

In the final reconstruction, we used 12 days of satellite retrievals. This information has now been added to Section 2.4 when introducing the final reconstruction. When multiple days of the satellite retrievals were used, all the earlier overpasses are considered. To clarify, we have updated these information in the revised text: "First, all TROPOMI overpasses over many days, in the final reconstruction 12 days, were used and second, an increased number of particles, 5 million air parcels, were released in the backward run."

Figure 9: Would it be possible to extent the figure to longer timescales? Based on figures 3, 10 and 13-15 the data is available, so I wonder why it wasn't included here? L270: Why did the authors choose to implement the constant injection rate to represent 1.5 Tg and not 2.1 Tg like the other simulations? Earlier it is established that 1.5 Tg does not give

realistic values, so is it considered for the constant injection rate. When looking at figure 9, moving up the constant emission simulation by 0.6 Tg, apart from the initial peak, you get a much better comparison with TROPOMI and the results for this simulation also fall within the satellite uncertainty range after several days, like the other two simulations. Also, it is not clear to me if the chemistry module is used for the constant injection rate simulation or whether it uses the 14-day exponential decay (similar to the orange line in figure 9). Based on how smooth the removal is in figure 9 for the purple line, I think the latter is true, but this needs to be clarified.

At first place, this project was focused on the early stage of the eruption. And the project was gradually extended to longer time ranges. We do have intermediate results extend at least to 15 July. And the part not shown in the paper is similar with the Figure 11 in (de Leeuw et al., 2021) and does not influence the interpretation and conclusions of the paper. Therefore, we did not extend Figure 9 to longer timescales. 1.5 Tg were used in earlier studies (Global Volcanism Program, 2019; Muser et al., 2020; de Leeuw et al., 2021) and it represents the maximum $SO_2$ mass retrieved on a single day. That's why we used it as the target to compare with. Using the 2.1 Tg for the constant injection rate, the forward simulation captures the mass change after June 28 but not before June 28. In the forward simulations, the mass were derived from the chemistry module rather than 14-day exponential decay. This has been emphasized in the revised figure caption.

L277: Which chemical reactions are included in the OH chemistry module and what are the reaction rates? I think a description of the chemistry module or a citation explaining the module should be included.

We added a reference to Hoffmann et al. (2021), providing a detailed description of the OH chemistry module.

L283: "... are consistent with the total $SO_2$ mass derived from TROPOMI estimations." If I interpret figure 8 correctly, it seems that the MPTRAC data in figure 9a are initialised using the TROPOMI retrievals during the first 10-12 days. Therefore, I wonder how independent the two datasets are for the period shown in the figure and whether the very good agreement is a direct consequence of the way how the MPTRAC simulations are initialised using the same retrievals to which it is now again compared against.

Indeed there is a dependency in initializing a forward simulation with TROPOMI data and then comparing the results to TROPOMI data. In Fig. 9a our approach checks the internal consistency between the estimated emissions, forward simulations, and the observations. However, in figure 9b, the simulations are initialised using the AIRS retrievals but the mass compared against is derived from TROPOMI. Figure 9a and 9b show similar performance, and at least for figure 9b, the two datasets are independent.

L300-305: The constant injection rate simulation assumes that the mass is emitted uniformly between 5 and 15 km altitude. Assuming a tropopause at 10 km, this means that approximately 50% of the mass is emitted into the troposphere, where the lifetime of $SO_2$ is much shorter (as seen in figure 10). If you would use a constant injection rate that has

a more realistic profile with more of the mass emitted into the lower stratosphere/upper troposphere, how much would this improve the comparison?

Yes, we agree with the reviewer that the mass is very dependent on the profile of the injection. In the early phase of this project, we initialized the simulations with the VolRes profile. However, as de Leeuw et al. (2021) already have done extensive tests using the VolRes profile and our results using the VolRes are similar with de Leeuw et al. (2021), we did not see an additional value in repeating these results here, but using rather a more simple "guess" of a constant injection rate. As shown in Figure 11 of de Leeuw et al. (2021), using a total injection of $1.5\,\mathrm{Tg}$ and the VolRes profile would result a similar problem as shown in our result using the constant injection rate. They suggested that either more $SO_2$ should be injected into the stratosphere ($1.09\,\mathrm{Tg}$) or a total of $2.0\,\mathrm{Tg}$ would be needed to match the TROPOMI observations. As the partitioning between stratospheric fraction (43%) and tropospheric fraction (57%) for the VolRes profile ($1.5\,\mathrm{Tg}$) is comparable to our reconstruction of 40.5% stratospheric fraction and 59.5% tropospheric fraction for $2.1\,\mathrm{Tg}$ total mass, we rather conclude that a higher total mass was more likely than the suggested increase of the stratospheric injection, which would result in 72% of the total mass in the stratosphere.

L327: "... mainly associated with transport of $SO_2$ in the lower troposphere (between 0 and 5km), which was not represented in both initializations". Why did the backward trajectory method not manage to track back the TROPOMI footprints to the volcano for this part of the plume? Based on figures 11a and b, most of the southern branch retrieved by TROPOMI is not present in the MPTRAC simulations. The fact that the method does not seem to capture this large area of the plume associated with the lower emission altitudes makes me doubt the robustness of the method (see point L179), especially for lower altitude eruptions? I think this potential limitation should be discussed in the discussion section.

The most possible reason would be that the column density of this part of the $SO_2$ cloud is far more lower than the main $SO_2$ cloud. "Comparing with the major northern branch $SO_2$ cloud, note that the southern branch is very weak with $SO_2$ column densities mostly less than $10\,\mathrm{DU}$." The following text has been added to the revised text: "The much lower column density reduces the chance of identifying a source associated with this part of the $SO_2$ cloud. In addition, the TROPOMI averaging kernel significantly reduces the air parcels started at altitudes below $5\,\mathrm{km}$ which further reduces the chance of identifying a source at altitudes lower than $5\,\mathrm{km}$."

L333: Would it be possible that the part not represented by the constant emission injection rate simulation in panel 11f is linked to emissions at lower altitudes that were also missing in panels 11b-c? I think it would be very useful to repeat the analysis for the VolRes profile (which includes the lower-level emissions) to see if this could explain part of the differences observed.

As mentioned in previous comments and responses, our earlier simulations using the VolRes profile are highly similar with the results shown in de Leeuw et al. (2021). Therefore, such

results are not repeatedly shown here. In the revised text, the following discussion is added: "Injections at lower altitudes from the VolRes profile could also partly explain the part not represented in Figure 11f, however that part is also underestimated in the forward simulation initialised by the VolRes profile de Leeuw et al. (2021)."

Fig. 13-14: I think these figures can be combined to one figure with 6 panels, rather than having two separate figures. Label of Fig.13: (CSI, a) → (CSI, c)

We have combined Figure 13 and Figure 14 into one figure.

Fig. 15: 'Color coding indicates the column density threshold...' This is incorrect, as the different colors show the different simulation initialisations at a constant detection threshold (5 DU).

There was a mistake in the figure caption. It has now been removed. And the following has been added to the figure caption: "The column density threshold used to detect events is $5.0\,DU$."

L410-414: I think the authors should include the altitudes used in MPTRAC for this part of the analysis. Maybe in a table in the supplementary material could be a good option, as the current manuscript is pretty long already.

The altitudes were derivied from CALIOP following Gorkavyi et al. (2021). In the revised manuscript, we have added information on the value of altitudes during the study period: "During 17 July to 21 July 2019, the aerosol altitude is $\sim18\,km$, and it rises to $\sim20\,km$ during 24-27 July, after which it gradually rises to $24\,km$ around 14 August 2019 (Gorkavyi et al., 2021)."

L460: I miss a short discussion of the limitations/uncertainties related to the MPTRAC model and the backward trajectory method. One potential impact not discussed is the impact of the lofting of the plume due to the co-existence of ash. In the paper by Muser et al. 2020, a lofting effect was identified for the Raikoke plume during the initial days after the eruption. I can't find any information that the MPTRAC model accounts for this effect and as a result the back-trajectories could be placed at the wrong altitudes in the reconstruction. This in turn could have an impact on the forward simulations on longer timescales.

In the revised text, we have added discussion on the impact of the lofting of the plume due to the co-existence of ash. The fowlling discussion has been added in the revised text: "Besides the above limitations, the current reconstruction and in turn the forward simulations may be also influenced by the selection of the TROPOMI products, i.e. the altitude of assumed $SO_2$ layer during retrieval, and by the lofting of the plume due to the co-existence of ash......" and "On the other hand, Muser et al. (2020) reported a lofting effect of ash for the Raikoke plume during the initial days after the eruption. The lofting effect may also exist during the period of the compact $SO_2$ (Gorkavyi et al., 2021). Such lofting effect would directly influence the forward simulation as it is not reflected in the meteorological data (Section 3.2.4) and may need manual turning to correctly simulate

the long-range transport of the $SO_2$. As the vertical column density was used in the reconstruction and it does not contain vertical information, the lofting effect may has less influence on the reconstruction. A quantitative assessment of the impact of the lofting effect is however unavailable from the current study, and it should be considered in the future study."

L474: The study by de Leeuw et al. shows that the 2.0 Tg emission profile overestimates the $SO_2$ mass from the TROPOMI retrievals during the first few days after the eruption. Therefore, I think this statement would be more accurate when adding 'to match the TROPOMI retrievals on timescales > 1 week'

Fixed

L475: Are the stratospheric amounts similar for all the simulations considered or is this the value for the most accurate simulation? If increasing the emission to 2.1 Tg for the constant emission case, how much would be emitted into the stratosphere? I think it would be a very useful addition if the authors could include an uncertainty range for the 0.85 Tg estimate using the full range of simulations they have performed.

The stratospheric amount is also different in simulations with different total injections. In the revised text, an uncertainty range for the 0.85 Tg estimate has been added. Now it is expressed as $0.85 \pm 0.08$ Tg.

L483: "From our forward simulations, the second and third plume are potentially overestimated." Would you be able to identify potential reasons for this overestimation? I think a short discussion what might have caused the overestimation should be included here.

In the revised text, we have added discussion on the potential reason: "In the current setting, the backward trajectory method would only pick the first hit to identify a new source and the second overpass will be not counted anymore. In realty, however, the $SO_2$ may have passed the volcano multiple times, which may lead to an overestimation for the second and the third plume."

L488 and L499: 2 weeks → 10 days. In this study only the location and spatial extend during first 10 days are discussed.

Corrected.

L518: I think it would be useful to include the fraction of the mass emitted into the stratosphere (0.85 Tg) here? Especially for climate impact studies, it is mainly the long-term stratospheric part of the plume that is of particular interest, as most of the tropospheric signal will be removed after several days/week.

We agree with the reviewer that the mass emitted into the stratosphere has a profound climate impact. We added the fraction of the mass emitted into the stratosphere in the revised text: "$40.5\%$ (0.85 Tg) of the total $SO_2$ mass were injected into the lower stratosphere."

L519: Better than what? The study does not show how the presented mass estimates compare with mass estimates using the profiles from other studies.

We mean better than using a constant emission of 1.5 Tg. In the revised text, we added a reference to Figure 9: "We consider our new estimation of a larger amount of $SO_2$ reasonable, as it better reproduces the retrieved mass change in forward simulations than assuming a constant emission of 1.5 Tg (Fig. 9)."

Technical corrections/suggestions:

Order of the figure panels. Some figures have panel b above a (fig. 3) and others have panel b below panel a (fig. 7). Please use one consistent ordering of labelling the panels in the figures to avoid confusion.

Fixed.

L40.: Besides → Beside

Fixed.

L41: parcels → parcel

Fixed.

L71: observed from satellite → retrieved by the satellite

Fixed.

L78: since begin of → since the beginning of

Fixed.

L85: satellite observations → satellite retrievals

Fixed.

L106: DU → Dobson Unit (DU)

Fixed.

L178: As both, AIRS and TROPOMI → As both AIRS and TROPOMI

Fixed.

L199: turned → tuned

Fixed.

Fig.7: Differences of → Differences in

Fixed.

Fig.7: TROPIMI → TROPOMI

Fixed.

L255: short term → short-term

Fixed.

L256: long term → long-term

Fixed.

References: Please check all the references carefully, as some have the DOI included twice (e.g. Hoffman et al 2014). Also use abbreviations for all the journals consistently.

Done.

**References**

de Leeuw, J., Schmidt, A., Witham, C. S., Theys, N., Taylor, I. A., Grainger, R. G., Pope, R. J., Haywood, J., Osborne, M., and Kristiansen, N. I.: The 2019 Raikoke volcanic eruption – Part 1: Dispersion model simulations and satellite retrievals of volcanic sulfur dioxide, Atmos. Chem. Phys., 21, 10 851–10 879, doi: 10.5194/acp-21-10851-2021, 2021.

Global Volcanism Program: Report on Raikoke (Russia), in: Bulletin of the Global Volcanism Network, edited by Crafford, A. and Venzke, E., vol. 44, doi: 10.5479/si.gvp.bgvn201908-290250, 2019.

Gorkavyi, N., Krotkov, N., Li, C., Lait, L., Colarco, P., Carn, S., DeLand, M., Newman, P., Schoeberl, M., Taha, G., Torres, O., Vasilkov, A., and Joiner, J.: Tracking aerosols and $SO_2$ clouds from the Raikoke eruption: 3D view from satellite observations, Atmos. Meas. Tech. Discuss., 2021, 1–22, doi: 10.5194/amt-2021-58, 2021.

Hoffmann, L., Rößler, T., Griessbach, S., Heng, Y., and Stein, O.: Lagrangian transport simulations of volcanic sulfur dioxide emissions: Impact of meteorological data products, J. Geophys. Res., 121, 4651–4673, doi: 10.1002/2015JD023749, 2016.

Hoffmann, L., Baumeister, P. F., Cai, Z., Clemens, J., Griessbach, S., Günther, G., Heng, Y., Liu, M., Haghighi Mood, K., Stein, O., Thomas, N., Vogel, B., Wu, X., and Zou, L.: Massive-Parallel Trajectory Calculations version 2.2 (MPTRAC-2.2): Lagrangian transport simulations on Graphics Processing Units (GPUs), Geosci. Model Dev. Discuss., 2021, 1–51, doi: 10.5194/gmd-2021-382, 2021.

Kristiansen, N. I., Stohl, A., Prata, A. J., Bukowiecki, N., Dacre, H., Eckhardt, S., Henne, S., Hort, M. C., Johnson, B. T., Marenco, F., Neininger, B., Reitebuch, O., Seibert, P., Thomson, D. J., Webster, H. N., and Weinzierl, B.: Performance assessment of a volcanic ash transport model mini-ensemble used for inverse modeling of the 2010 Eyjafjallajökull eruption, Journal of Geophysical Research: Atmospheres, 117, doi: https://doi.org/10.1029/2011JD016844, URL

https://agupubs.onlinelibrary.wiley.com/doi/abs/10.1029/2011JD016844, 2012.

Muser, L. O., Hoshyaripour, G. A., Bruckert, J., Horváth, A., Malinina, E., Wallis, S., Prata, F. J., Rozanov, A., von Savigny, C., Vogel, H., and Vogel, B.: Particle aging and aerosol–radiation interaction affect volcanic plume dispersion: evidence from the Raikoke 2019 eruption, Atmos. Chem. Phys., 20, 15 015–15 036, doi: 10.5194/acp-20-15015-2020, 2020.

Pommrich, R., Müller, R., Grooß, J.-U., Konopka, P., Ploeger, F., Vogel, B., Tao, M., Hoppe, C. M., Günther, G., Spelten, N., Hoffmann, L., Pumphrey, H.-C., Viciani, S., D'Amato, F., Volk, C. M., Hoor, P., Schlager, H., and Riese, M.: Tropical troposphere to stratosphere transport of carbon monoxide and long-lived trace species in the Chemical Lagrangian Model of the Stratosphere (CLaMS), Geophys. Mod. Dev., 7, 2895–2916, doi: 10.5194/gmd-7-2895-2014, 2014.

Prata, A. J., Gangale, G., Clarisse, L., and Karagulian, F.: Ash and sulfur dioxide in the 2008 eruptions of Okmok and Kasatochi: Insights from high spectral resolution satellite measurements, Journal of Geophysical Research: Atmospheres, 115, doi: https://doi.org/10.1029/2009JD013556, URL https://agupubs.onlinelibrary.wiley.com/doi/abs/10.1029/2009JD013556, 2010.

WMO: Meteorology A Three-Dimensional Science: Second Session of the Commission for Aerology, WMO Bull., iv, 134–138, 1957.

---

## Author Response (AR1)

**Reply to review comments**

We thank the reviewers for the time and effort spent on the manuscript and for providing helpful comments. We considered all comments and hope that the revised draft properly addresses the open issues. Please find our point-by-point replies below (colored in blue).

**Reviewer #1**

General comments:

Cai et al. describe an improved method to derive realistic time- and altitude-resolved volcanic SO2 emission rates based on satellite observations of SO2 and Lagrangian backward simulations. This study is the first application of the upgraded MPTRAC model in inverse modeling of volcanic SO2 injections and transport. By considering varieties of SO2 observations, adjusting the SO2 mass and initialization method, and including an OH chemical module, the updated procedure provided a more sophisticated way to retrieve explosive volcanic SO2 emission and achieved more promising results than their earlier version. The calibrations and sensitive tests make this study also a good tool article for the use of the MPTRAC model. The method described is of great potential in refining volcanic SO2 emissions in climate models. This manuscript is clearly organized and well written, so I recommend this study be published after minor revision.

We thank the reviewer for the overall positive comments and have revised the manuscript according to the comments and suggestions provided by the reviewer. Please see a detailed reply to each comment below on the revisions in the revised manuscript.

My specific comments are as follows:

L1 are important
Corrected.
L10 The reconstructed SO2 injection...
Added "SO2".
L71 observed from the satellite
Changed into "retrieved by the satellite".
L78 since the beginning of operations
Corrected.

L89 compact  $SO_2$  clouds

Corrected.

L106 5 Dobson Unit (DU)

Fixed.

Figure 1 caption: middle latitudes or mid-latitudes

Changed into "mid-latitudes".

Figure 2: 24-hour

Corrected.

Figure 2: Are there gaps between tracks of AIRS and TROPOMI on the demonstrated area? If so, it would be easier for readers to distinguish between track gaps and areas with little  $SO_2$  with the track gaps indicated.

For TROPOMI, there are no gaps between tracks on the demonstrated area. For AIRS, only minor gaps exist between tracks in the region of 40-50N. However, the  $SO_2$  cloud was not located in the gaps between tracks on the selected day. Therefore, the tracks were not shown in the figure.

L142 and hereafter, subgrid-scale

Changed throughout the manuscript.

Does the thick black solid line indicate the altitude of the tropopause?

Yes, it is the altitude of the tropopause calculated based on the WMO temperature lapserate definition. This information has been added in the revised figure caption.

Figure 7: Although the cumulative  $SO_2$  emissions (Fig.5) from TROPOMI and AIRS nighttime are similar, the magnitude of the differences in emission rates in Fig. 7(a) seems almost as large as the emission rate in Fig. 3(a). Based on your current results, could you conclude the best practices (including a suite of satellite data and exterior winds) for the Raikoke case?

We cannot give a recommendation on the best choice of satellite and meteorological data to be used for estimating the  $SO_2$  injections. However, our results indicate that the cumulative masses are robust, irrespective of the selected satellite and wind data set. The significant

differences in Fig. 7 mainly arise from differences in the timing of the reconstructed injections. Our recommendation would be to conduct tests with different data sets to assess their impact on the results. Such tests have been done in the current paper as well as previously by Hoffmann et al. (2016) and Kristiansen et al. (2012).

L255–256 short-term, long-term, longer-term

Changed throughout the manuscript.

L277 Would you please specify the source data of hydroxyl radical or how the profile of hydroxyl radical is set for the OH chemistry module in the model?

To clarify, we added: "Monthly mean zonal mean OH concentrations are obtained from the study of Pommrich et al. (2014)."

Figure 15: I guess there is a mistake in the figure caption. Figure 15 shows the POD, FAR, and CSI time series of forward simulations initialized with TROPOMI observations, AIRS nighttime observations, and a constant injection rate when the detection threshold was set to 5.0 DU.

Yes, there was a mistake in the figure caption. "Color coding indicates the column density threshold used to detect events (see plot key)." has been removed, and "The column density threshold used to detect events is 5.0 DU." has been added.

L410–414: Did you fix the  $SO_2$  column in a certain altitude level constrained by the altitudes of aerosols provided by CALIOP? Please make the altitude range clear, so readers do not have to search for and read Gorkavyi et al. only for the value of altitudes.

Yes, the SO2 column are fixed to a certain altitude level constrained by the altitudes of aerosols provided by CALIOP. In the revised manuscript, we have added information on the value of altitudes during the study period: "During 17 July to 21 July 2019, the aerosol altitude is  $\sim 18$  km, and it rises to  $\sim 20$  km during 24-27 July, after which it gradually rises to 24 km around 14 August 2019 (Gorkavyi et al., 2021)."

L429 a constant potential temperature level/a isentropic surface.

Corrected.

**Others**

Please double-check the use of en dashes and hyphens between ranges of number and dates.

It is fixed now.

**Reviewer #2**

General comments:

The study by Cai et al. combines satellite retrievals from TROPOMI and AIRS with the Lagrangian transport model MPTRAC to give a detailed estimate of the volcanic  $SO_2$  injections for the 2019 Raikoke eruption. By applying an inverse modelling technique, the authors give a detailed estimate of the time- and altitude-resolved  $SO_2$  emission for this eruption. This study investigated a range of conditions in the latest version of MPTRAC (e.g., using various  $SO_2$  retrievals, varying diffusion parameters, scaling of the initial  $SO_2$  mass, including a new OH chemistry module), which results in a detailed consideration of the estimated  $SO_2$  injections.

Initialising forward trajectories with this new  $SO_2$  estimates for Raikoke can reproduce the  $SO_2$  mass variations and spatial dispersion of the volcanic cloud retrieved by TROPOMI during the first 10 days after the eruption. Finally, the impact of diffusion is investigated, showing that it is too strong and as a result the model cannot capture the internal structure of the simulated  $SO_2$  cloud well.

This work is very relevant to the atmospheric modelling community. Detailed eruption source parameters for volcanic eruptions are known to be difficult to determine. This study is a useful addition to the existing literature demonstrating that inverse modelling techniques are very useful to create more detailed eruption source parameters to initialise model simulations.

The manuscript is rather long, but I cannot see how to make it much shorter other than merging some of the figures. It is well written, and the figures are of a good quality and the authors give clear interpretations of the data. I have some minor concerns about some of the methodology used in this study, as outlined in the specific comments below. However, the overall work presented in the manuscript is good. I therefore recommend minor revisions to address the points outlined below before publication.

We thank the reviewer for the constructive comments and the detailed suggestions on the manuscript. We have studied each of the comments and suggestions and revised the manuscript accordingly. The discussion on the limitations and uncertainties associated with the current study is strengthened in the revised manuscript. Please see below the replies to each comment/suggestion and the corresponding revisions in the revised manuscript.

Specific comments:

L11: Satellites do not directly observe  $SO_2$ , but (as explained in sections 2.1 and 2.2) uses various bands in the infrared/UV spectra to retrieve estimates for  $SO_2$ . Therefore, in general it is better to use the terms 'retrievals/retrieved', rather than 'observations/observed' when discussing the satellite products. Please check carefully throughout the manuscript.

Following the suggestion, we have checked the use of "observations/observed" throughout

the manuscript and have changed them into "retrievals/retrieved".

L114: "The AIRS... upper levels." I think this sentence needs a reference that supports this statement.

We found that Prata et al. (2010) pointed out that the effect of scattered solar radiation on the 7.3  $\mu$ m waveband of SO2 is negligible. Therefore, we removed this sentence with our hypothesis from the paper.

L115: 'particularly carefully'. Please avoid non-scientific terms. It should be clarified how the authors used the daytime data.

We would like to stress that AIRS day- and nighttime data should not be mixed up in the analysis. We shortened the sentence to "Therefore, the AIRS nighttime and daytime data are considered separately in this study.", to clarify.

L130: In this study the results from the 15 km retrieval are presented. However, based on Fig. 3, you could argue that the assumed SO2 height for the 15 km retrieval product is too high in the atmosphere for this eruption and that the 7 km retrieval from TROPOMI could be considered equally realistic (especially for the second and third phase). Did the authors investigate the impact of using the 7 km retrieval on their results? The 15 km and 7 km retrieval products for TROPOMI can result in different SO2 column mass estimates, which would in turn could also influence your estimate of the total emitted mass. It would be very interesting to understand if you would still get the reported  $2.1\pm0.2$ Tg estimate when applying the 7 km retrieval. I think some discussion on this potential source of uncertainty should be included in the paper.

We have previously looked at the different TROPOMI  $SO_2$  products that assume the  $SO_2$ layer is at either 1, 7, or 15 km above sea level. Although the different retrievals assumed different  $SO_2$  layer height, the vertical column density product itself does not contain altitude information. The main difference between different products is the absolute value of the vertical column density. Therefore, it does not influence the reconstruction of the relative injection rate. But when calibrating the absolute  $SO_2$  mass (section 3.1.2), the different product could have an impact due to difference in the absolute  $SO_2$  column density. Comparison of the total  $SO_2$  mass between the 7 km and 15 km retrieval products for TROPOMI shows that the mass is identical during the first week of the eruption and then the 7 km product gets an overall higher mass estimate. After the first week, the mass derived from the 7 km product is consistently higher than 15 km product by  $\sim 10$  percent. Therefore, using the 7 km product would get an higher estimate of the total  $SO_2$  mass but it is still within our reported uncertainty range. The following text has been added to the revised Discussion section: "Besides the above limitations, the current reconstruction and in turn the forward simulations may be also influenced by the selection of the TROPOMI products, i.e. the altitude of assumed  $SO_2$  layer during retrieval, and by the lofting of the plume due to the co-existence of ash. TROPOMI SO2 products are available for different scenarios that assume the  $SO_2$  is at either 1, 7, or 15 km above sea level. The main difference between different products is the absolute value of the vertical column density, and it has minor influence on the reconstruction of the relative injection rate. However, the different  $SO_2$  can result in different mass estimate. Comparison of the total  $SO_2$  mass between the 7 km and 15 km retrieval products for TROPOMI show that the mass is identical during the first week of the eruption. After the first week, the mass derived from the 7 km product is consistently higher than the 15 km product by ~10 percent. Therefore, using the 7 km product would get an higher estimate of the total  $SO_2$  mass, which is at the upper limit of the estimate reported in this study."

L155: "... reaction with the hydroxyl radical (OH)." What data is used to provide the OH field for the MPTRAC simulations?

To clarify, we added: "Monthly mean zonal mean OH concentrations are obtained from the study of Pommrich et al. (2014)."

L179-190: The work presented here samples trajectories from each satellite footprint between 0-25 km and combine them to obtain a best estimate of the SO2 emission. However, how certain is it that there is a single best solution to the problem? Is it possible that a different emission profile not found by this method could give an equally good comparison with the TROPOMI retrievals? For example, assume two hypothetical layers (say at 2 km and 7 km) at a given TROPOMI pixel location that both can be retraced to the volcano. I am not sure I understand how the backward trajectory method is able to determine which layer contained the SO2 mass (or is able to determine the ratio between the two layers)? If I understand correctly, based on the sensitivity of the TROPOMI satellite (fig. 1), the method seems to be biased towards the higher altitudes, as more trajectories are released from the higher layers (7 km in our example case). But in our example, it is also possible that the mass was emitted in the lower layer, as it would give the same column total mass in TROPOMI. Is the backward trajectory method able exclude either possibility?

Yes, it is possible that for a given TROPOMI pixel location there may be multiple solutions in the backward trajectory method, especially if the wind does not change with height and over time. Taking your example and assuming that the wind at 2 and 7 km is the same, but different in-between. In the first step at all altitudes the same number of air parcels is initialized. Applying the TROPOMI kernel function redistributes the air parcels such, that about twice as much air parcels would start at 7 km compared to 2 km. MPTRAC would trace the air parcels at 2 and 7 km back to the volcano. So, in this case there is the hypothetical chance to falsely attribute a large  $SO_2$  mass at 2 km to 7 km, if only the layer at 2 km contains the volcanic plume. Fortunately the wind changes with height and time. In the study, this uncertainty has been reduced through two ways. First, TROPOMI retrievals over many days, in the final reconstruction 12 days, were used and second, an increased number of particles, 5 million air parcels, were released in the backward run. To make this clearer we added the aforementioned discussion to section 2.4.

L193: "fixed e-folding lifetime". How realistic is it to use a constant e-folding lifetime for the entire altitude range considered? As mentioned by the authors (L.296-302, Fig.10), the lifetime of  $SO_2$  varies strongly between the troposphere and the stratosphere. How should

we interpret this fixed e-folding lifetime? Is it an altitude-weighted average lifetime for the  $SO_2$  ?

The e-folding lifetime assumed here is to represent the overall removal of the  $SO_2$  in the atmosphere which differs from the lifetime of  $SO_2$  at certain altitude levels. The value was empirically determined here and more detailed information on it should refer to later sections of the paper (including also Fig. 10). We have revised the text in the manuscript to clarify.

L199: "To achieve a total injection of 2.1 Tg." So far, no evidence is given why 2.1 Tg would be more realistic. I think the authors should point to the discussion presented in section 3.1.2 or give a short explanation here on why the results have been tuned to 2.1 Tg. Fig.3: What does the black line in panel 3a represent?

Following the reviewer's suggestion, we have added a forward reference to Section 3.1.2 in the revised text. The black line in panel 3a (it is now 3b in the revised manuscript) represents the ERA5 temperature lapse-rate tropopause over Raikoke volcano. We have added this information to the revised figure caption.

L202: "... prominent second and third plume..." Looking at the video's provided in the study by de Leeuw et al, their dispersion simulations show that part of the first plume is dispersed back to the location of the Raikoke volcano around the same time this study identified the third plume (27-28 June). This is also shown by the red trajectory in figure 11g. How would the backward trajectory methodology deal which such an event and is it possible that this third peak in the shown analysis is partly a reoccurrence of the first plume at the Raikoke location. If back trajectories hit the source location multiple times, would it pick the first hit only, identifying a 'new' source, or would it consider the possibility of multiple overpasses over the volcano at earlier times? Related to this point, the video for  $SO_2$  dispersion provided in the study by de Leeuw et al. also shows that part of the  $SO_2$ cloud (using the VolRes profile) initially moves into a North-West direction, followed by it moving back and across the Raikoke volcano location around the 25 th of June. Based Fig.11 in the current paper, this part of the plume is not present in the presented dispersion experiment (panel 11b), while it is visible on the TROPOMI retrieval (panel 11a). This suggests that it might be linked to emissions at low altitudes (

Figure R1: HIMAWARI true color image on 27.06.2019 00:10 UTC showing a faint plume downstream of Raikoke.

 $^1\mathrm{HIMAWARI}$  data was downloaded from the Data Integration and Analysis System (DIAS) by the University of Tokyo.

In the earlier phase of this project, we also tried to use the VolRes profile. However, as de Leeuw et al. (2021) already done extensive tests using the VolRes profile and our results using the VolRes profile are similar with de Leeuw et al. (2021), we did not show the results using VolRes profile, but using rather a more simple "guess" of a constant injection rate. In the revised text, we have added results from simulations using the VolRes profile in terms of total SO2 mass (Fig. 9) and critical success index assessment (Fig. 14, previously the Fig. 15 before revision). The simulations using our reconstructions perform better in terms of mass and probability of detection (POD). As the Fig. 11 already takes quite a large space and SO2 distributions in the forward simulation using VolRes profile in this study again. In the end, our results are similar to the VolRes profile (Figure 4), despite the approach and data are rather different. In the revised text, we have strengthened the discussion on the uncertainty regarding the second and third plume and the forward simulations using the VolRes profile shown by de Leeuw et al. (2021).

**L203: How is the tropopause defined?**

We applied the thermal lapse-rate tropopause definition of WMO (1957). This information has been added to the text. Additional information on the ERA5 tropopause data set applied here has been added to the code and data availability section of the paper. The sentence reads now: "The first plume crossed the temperature lapse-rate tropopause (WMO, 1957) and injected SO2 ..."

L248: "...it matters how many days of satellite observations are used...". How many days are used for the results presented in this study (e.g. figure 3)? I think this should be specified in the manuscript. I also think some additional explanation is missing that describes how multiple days of the TROPOMI satellite retrievals are combined to reconstruct the SO2 injections. When using 12 days of retrievals, does this mean that all the earlier overpasses are still considered? Or does this mean that the back-trajectories are calculated for 12 days to reconstruct the signal for this specific overpass?

In the final reconstruction, we used 12 days of satellite retrievals. This information has now been added to Section 2.4 when introducing the final reconstruction. When multiple days of the satellite retrievals were used, all the earlier overpasses are considered. To clarify, we have updated these information in the revised text: "First, all TROPOMI overpasses over many days, in the final reconstruction 12 days, were used and second, an increased number of particles, 5 million air parcels, were released in the backward run."

Figure 9: Would it be possible to extent the figure to longer timescales? Based on figures 3, 10 and 13-15 the data is available, so I wonder why it wasn't included here? L270: Why did the authors choose to implement the constant injection rate to represent 1.5 Tg and not 2.1 Tg like the other simulations? Earlier it is established that 1.5 Tg does not give realistic values, so is it considered for the constant injection rate. When looking at figure 9, moving up the constant emission simulation by 0.6 Tg, apart from the initial peak, you

get a much better comparison with TROPOMI and the results for this simulation also fall within the satellite uncertainty range after several days, like the other two simulations. Also, it is not clear to me if the chemistry module is used for the constant injection rate simulation or whether it uses the 14-day exponential decay (similar to the orange line in figure 9). Based on how smooth the removal is in figure 9 for the purple line, I think the latter is true, but this needs to be clarified.

Now the figure (as well as Fig. 6) has been extended to 13 July, which does not influence the interpretation and conclusions of the paper. At first place, this project was focused on the early stage of the eruption and the project was gradually extended to longer time ranges. The 1.5 Tg SO2 were used in earlier studies (Global Volcanism Program, 2019; Muser et al., 2020; de Leeuw et al., 2021) and it represents the maximum SO2 mass retrieved on a single day. That's the motivation for using it as the target to compare with. When using the 2.1 Tg for the constant injection rate, the forward simulation captures the mass change after June 28, but not before June 28. In the forward simulations, the mass was derived from the chemistry module rather than the 15-day (there was a typo in the earlier version of the manuscript, and it has now been corrected) exponential decay. This has been emphasized in the revised figure caption.

L277: Which chemical reactions are included in the OH chemistry module and what are the reaction rates? I think a description of the chemistry module or a citation explaining the module should be included.

We added a reference to Hoffmann et al. (2022), providing a detailed description of the OH chemistry module.

L283: "... are consistent with the total  $SO_2$  mass derived from TROPOMI estimations." If I interpret figure 8 correctly, it seems that the MPTRAC data in figure 9a are initialised using the TROPOMI retrievals during the first 10-12 days. Therefore, I wonder how independent the two datasets are for the period shown in the figure and whether the very good agreement is a direct consequence of the way how the MPTRAC simulations are initialised using the same retrievals to which it is now again compared against.

Indeed there is a dependency in initializing a forward simulation with TROPOMI data and then comparing the results to TROPOMI data. In Fig. 9a our approach checks the internal consistency between the estimated injections, forward simulations, and the observations. However, in Fig. 9b, the simulations are initialised using the AIRS retrievals but the mass compared against is derived from TROPOMI. Figure 9a and 9b show similar performance, and at least for Fig. 9b, the two datasets are independent.

L300-305: The constant injection rate simulation assumes that the mass is emitted uniformly between 5 and 15 km altitude. Assuming a tropopause at 10 km, this means that approximately 50% of the mass is emitted into the troposphere, where the lifetime of  $SO_2$  is much shorter (as seen in figure 10). If you would use a constant injection rate that has a more realistic profile with more of the mass emitted into the lower stratosphere/upper troposphere, how much would this improve the comparison?

Yes, we agree with the reviewer that the mass is very dependent on the profile of the injection. In the early phase of this project, we initialized the simulations with the VolRes profile. However, as de Leeuw et al. (2021) have already done extensive tests using the VolRes profile and our results using the VolRes profile are similar with de Leeuw et al. (2021), we did not see an additional value in repeating these results here, but using rather a more simple "guess" of a constant injection rate. Our forward simulation using the VolRes profile with a total injection of 1.5 Tg is now shown in the revised Fig. 9. Simulated SO2 mass variation using the VolRes profile is very similar to the result using the constant injection rate. de Leeuw et al. (2021) suggested that either more SO2 should be injected into the stratosphere (1.09 Tg) or a total of 2.0 Tg would be needed to match the TROPOMI observations. As the partitioning between stratospheric fraction (43%) and tropospheric fraction (57%) for the VolRes profile (1.5 Tg) is comparable to our reconstruction of 40.5% stratospheric fraction and 59.5% tropospheric fraction for 2.1 Tg total mass, we rather conclude that a higher total mass was more likely than the suggested increase of the stratospheric injection, which would result in 72% of the total mass in the stratosphere.

L327: "... mainly associated with transport of  $SO_2$  in the lower troposphere (between 0 and 5km), which was not represented in both initializations". Why did the backward trajectory method not manage to track back the TROPOMI footprints to the volcano for this part of the plume? Based on figures 11a and b, most of the southern branch retrieved by TROPOMI is not present in the MPTRAC simulations. The fact that the method does not seem to capture this large area of the plume associated with the lower emission altitudes makes me doubt the robustness of the method (see point L179), especially for lower altitude eruptions? I think this potential limitation should be discussed in the discussion section.

The most possible reason would be that the column density of this part of the  $SO_2$  cloud is far more lower than the main  $SO_2$  cloud. "Comparing with the major northern branch  $SO_2$  cloud, note that the southern branch is very weak with  $SO_2$  column densities mostly less than 10 DU." The following text has been added to the revised text: "The much lower column density reduces the chance of identifying a source associated with this part of the  $SO_2$  cloud. In addition, the TROPOMI averaging kernel significantly reduces the air parcels started at altitudes below 5 km, which further reduces the chance of identifying a source at altitudes lower than 5 km."

L333: Would it be possible that the part not represented by the constant emission injection rate simulation in panel 11f is linked to emissions at lower altitudes that were also missing in panels 11b-c? I think it would be very useful to repeat the analysis for the VolRes profile (which includes the lower-level emissions) to see if this could explain part of the differences observed.

As mentioned in previous comments and responses, our earlier simulations using the VolRes profile are highly similar with the results shown in de Leeuw et al. (2021). Therefore, such results are not repeatedly shown here. In the revised text, the following discussion is added: "Injections at lower altitudes from the VolRes profile could also partly explain the part not represented in Figure 11f, however that part is also underestimated in the forward

simulation initialised by the VolRes profile shown by de Leeuw et al. (2021)."

Fig. 13-14: I think these figures can be combined to one figure with 6 panels, rather than having two separate figures. Label of Fig.13: (CSI, a)  $\rightarrow$  (CSI, c)

We have combined Fig. 13 and Fig. 14 into one figure.

Fig. 15: 'Color coding indicates the column density threshold...' This is incorrect, as the different colors show the different simulation initialisations at a constant detection threshold (5 DU).

There was a mistake in the figure caption. It has now been removed. And the following has been added to the figure caption: "The column density threshold used to detect events is 5.0 DU."

L410-414: I think the authors should include the altitudes used in MPTRAC for this part of the analysis. Maybe in a table in the supplementary material could be a good option, as the current manuscript is pretty long already.

The altitudes were derived from CALIOP following Gorkavyi et al. (2021). In the revised manuscript, we have added information on the value of altitudes during the study period: "During 17 July to 21 July 2019, the aerosol altitude is  $\sim 18$  km, and it rises to  $\sim 20$  km during 24-27 July, after which it gradually rises to 24 km around 14 August 2019 (Gorkavyi et al., 2021)."

L460: I miss a short discussion of the limitations/uncertainties related to the MPTRAC model and the backward trajectory method. One potential impact not discussed is the impact of the lofting of the plume due to the co-existence of ash. In the paper by Muser et al. 2020, a lofting effect was identified for the Raikoke plume during the initial days after the eruption. I can't find any information that the MPTRAC model accounts for this effect and as a result the back-trajectories could be placed at the wrong altitudes in the reconstruction. This in turn could have an impact on the forward simulations on longer timescales.

In the revised text, we have added a discussion on the impact of the lofting of the plume due to the co-existence of ash. The following discussion has been added in the revised Discussion section: "Besides the above limitations, the current reconstruction and in turn the forward simulations may be also influenced by the selection of the TROPOMI products, i.e. the altitude of assumed SO2 layer during retrieval, and by the lofting of the plume due to the co-existence of ash....." and "On the other hand, Muser et al. (2020) reported a lofting effect of ash for the Raikoke plume during the initial days after the eruption. The lofting effect may also exist during the period of the compact SO2 (Gorkavyi et al., 2021). Such lofting effect would directly influence the forward simulation as it is not reflected in the meteorological data (Section 3.2.4) and may need manual tuning to correctly simulate the long-range transport of the SO2. As the vertical column density was used in the reconstruction and it does not contain vertical information, the lofting effect may have less influence on the reconstruction. A quantitative assessment of the impact of the lofting effect is however unavailable from the current study, and it should be considered in the future study."

L474: The study by de Leeuw et al. shows that the 2.0 Tg emission profile overestimates the  $SO_2$  mass from the TROPOMI retrievals during the first few days after the eruption. Therefore, I think this statement would be more accurate when adding 'to match the TROPOMI retrievals on timescales > 1 week'

**Fixed**

L475: Are the stratospheric amounts similar for all the simulations considered or is this the value for the most accurate simulation? If increasing the emission to 2.1 Tg for the constant emission case, how much would be emitted into the stratosphere? I think it would be a very useful addition if the authors could include an uncertainty range for the 0.85 Tg estimate using the full range of simulations they have performed.

The stratospheric amount is also different in simulations with different total injections. In the revised text, an uncertainty range for the 0.85 Tg estimate has been added. Now it is expressed as  $0.85 \pm 0.08$  Tg.

L483: "From our forward simulations, the second and third plume are potentially overestimated." Would you be able to identify potential reasons for this overestimation? I think a short discussion what might have caused the overestimation should be included here.

In the revised text, we have added a discussion on the potential reason: "In the current setting, the backward trajectory method would only pick the first hit to identify a new source and the second overpass will be not counted anymore. In reality, however, the  $SO_2$  may have passed the volcano multiple times, which may lead to an overestimation for the second and the third plume."

L488 and L499: 2 weeks  $\rightarrow$  10 days. In this study only the location and spatial extend during first 10 days are discussed.

**Corrected.**

L518: I think it would be useful to include the fraction of the mass emitted into the stratosphere (0.85 Tg) here? Especially for climate impact studies, it is mainly the long-term stratospheric part of the plume that is of particular interest, as most of the tropospheric signal will be removed after several days/week.

We agree with the reviewer that the mass emitted into the stratosphere has a profound climate impact. We added the fraction of the mass emitted into the stratosphere in the revised text: "40.5% (0.85 Tg) of the total SO2 mass were injected into the lower stratosphere."

L519: Better than what? The study does not show how the presented mass estimates compare with mass estimates using the profiles from other studies.

Mass changes in the simulation using the VolRes profile has now been added in the revised

manuscript (Fig. 9). In the revised text, we added a reference to Fig. 9: "We consider our new estimation of a larger amount of  $SO_2$  reasonable, as it better reproduces the retrieved mass change in the forward simulations than assuming an injection of 1.5 Tg  $SO_2$  either by a constant injection rate or following the VolRes profile (Fig. 9)"

Technical corrections/suggestions:

Order of the figure panels. Some figures have panel b above a (fig. 3) and others have panel b below panel a (fig. 7). Please use one consistent ordering of labelling the panels in the figures to avoid confusion.

```
Fixed.
```

L40.: Besides  $\rightarrow$  Beside

Fixed.

L41: parcels  $\rightarrow$  parcel

Fixed.

L71: observed from satellite  $\rightarrow$  retrieved by the satellite

Fixed.

L78: since begin of  $\rightarrow$  since the beginning of

Fixed.

L85: satellite observations  $\rightarrow$  satellite retrievals

Fixed.

L106:  $DU \rightarrow Dobson Unit (DU)$

Fixed.

L178: As both, AIRS and TROPOMI  $\rightarrow$  As both AIRS and TROPOMI

Fixed.

L199: turned  $\rightarrow$  tuned

Fixed.

Fig.7: Differences of  $\rightarrow$  Differences in

Fixed.

Fig.7: TROPIMI  $\rightarrow$  TROPOMI

Fixed.

L255: short term  $\rightarrow$  short-term

Fixed.

L256: long term  $\rightarrow$  long-term

**Fixed.**

References: Please check all the references carefully, as some have the DOI included twice (e.g. Hoffman et al 2014). Also use abbreviations for all the journals consistently.

**Done.**

**References**

- de Leeuw, J., Schmidt, A., Witham, C. S., Theys, N., Taylor, I. A., Grainger, R. G., Pope, R. J., Haywood, J., Osborne, M., and Kristiansen, N. I.: The 2019 Raikoke volcanic eruption – Part 1: Dispersion model simulations and satellite retrievals of volcanic sulfur dioxide, Atmos. Chem. Phys., 21, 10851–10879, doi: 10.5194/acp-21-10851-2021, 2021.
- Global Volcanism Program: Report on Raikoke (Russia), in: Bulletin of the Global Volcanism Network, edited by Crafford, A. and Venzke, E., vol. 44, doi: 10.5479/si.gvp.bgvn201908-290250, 2019.
- Gorkavyi, N., Krotkov, N., Li, C., Lait, L., Colarco, P., Carn, S., DeLand, M., Newman, P., Schoeberl, M., Taha, G., Torres, O., Vasilkov, A., and Joiner, J.: Tracking aerosols and SO2 clouds from the Raikoke eruption: 3D view from satellite observations, Atmos. Meas. Tech. Discuss., 2021, 1–22, doi: 10.5194/amt-2021-58, 2021.
- Hoffmann, L., Rößler, T., Griessbach, S., Heng, Y., and Stein, O.: Lagrangian transport simulations of volcanic sulfur dioxide emissions: Impact of meteorological data products, J. Geophys. Res., 121, 4651–4673, doi: 10.1002/2015JD023749, 2016.
- Hoffmann, L., Baumeister, P. F., Cai, Z., Clemens, J., Griessbach, S., Günther, G., Heng, Y., Liu, M., Haghighi Mood, K., Stein, O., Thomas, N., Vogel, B., Wu, X., and Zou, L.: Massive-Parallel Trajectory Calculations version 2.2 (MPTRAC-2.2): Lagrangian transport simulations on Graphics Processing Units (GPUs), Geophys. Mod. Dev., 15, 2731–2762, doi: 10.5194/gmd-15-2731-2022, 2022.
- Kristiansen, N. I., Stohl, A., Prata, A. J., Bukowiecki, N., Dacre, H., Eckhardt, S., Henne, S., Hort, M. C., Johnson, B. T., Marenco, F., Neininger, B., Reitebuch, O., Seibert, P., Thomson, D. J., Webster, H. N., and Weinzierl, B.: Performance assessment of a volcanic ash transport model mini-ensemble used for inverse modeling of the 2010 Eyjafjallajökull eruption, Journal of Geophysical Research: Atmospheres, 117, doi: https://doi.org/10.1029/2011JD016844, 2012.
- Muser, L. O., Hoshyaripour, G. A., Bruckert, J., Horváth, A., Malinina, E., Wallis, S., Prata, F. J., Rozanov, A., von Savigny, C., Vogel, H., and Vogel, B.: Particle aging and

aerosol–radiation interaction affect volcanic plume dispersion: evidence from the Raikoke 2019 eruption, Atmos. Chem. Phys., 20,  $15\,015-15\,036$ , doi: 10.5194/acp-20-15015-2020, 2020.

- Pommrich, R., Müller, R., Grooß, J.-U., Konopka, P., Ploeger, F., Vogel, B., Tao, M., Hoppe, C. M., Günther, G., Spelten, N., Hoffmann, L., Pumphrey, H.-C., Viciani, S., D'Amato, F., Volk, C. M., Hoor, P., Schlager, H., and Riese, M.: Tropical troposphere to stratosphere transport of carbon monoxide and long-lived trace species in the Chemical Lagrangian Model of the Stratosphere (CLaMS), Geophys. Mod. Dev., 7, 2895–2916, doi: 10.5194/gmd-7-2895-2014, 2014.
- Prata, A. J., Gangale, G., Clarisse, L., and Karagulian, F.: Ash and sulfur dioxide in the 2008 eruptions of Okmok and Kasatochi: Insights from high spectral resolution satellite measurements, Journal of Geophysical Research: Atmospheres, 115, doi: https://doi.org/10.1029/2009JD013556, 2010.
- WMO: Meteorology A Three-Dimensional Science: Second Session of the Commission for Aerology, WMO Bull., iv, 134–138, 1957.

---

## Editor Decision (ED1)

Editor Comments on Manuscript No ACP-2021-874

P1, L2: applied -> apply

P2, L36: "…..both day- and nighttime near global coverage" does not sound correct. Please rephrase. Maybe you could write "near global coverage during both, day- and nighttime".

P2, L44: lagranto -> LAGRANTO

P3, L65: "from different studies".  This text part does not seem to be at the right place in the sentence. I think you should rather write: For instance the estimation of SO2 mass from the 2009 Sarychev eruption from different studies varies from 0.8 to 1.8 Tg (Fromm et al., 2014)."

P3, L89: "Finally, we discuss the results from our work comparing with previous studies in Sect.4 ……". Sentence not correct. I would suggest to write it as follows: " Finally, in Sect.4 we discuss the results from our work by comparing to previous studies………."

P4, L94: used -> use

P7, L164: add "a" and  "vertical" so that it reads: "….at a 31 km horizontal resolution and 137 vertical levels spanning from the surface up to 0.01 hPa."

P7, L165: interpolated to 0.3 x 0.3 horizonzal resolution -> inperpolated to a 0.3 x 0.3 horizontal resolution

P7, L177: of the SO2 -> of SO2

P8, L212. Delete "together"

P9, L219: troposphere -> tropospheric

P10, Fig 4 caption: Better to write "plot legend" than "plot key"?

P10, 224: Especially here the usage of retrieval is quite confusing. This paragraph should be reworked.

P11, Fig 5 caption. Add "the" -> "…. of the Raikoke SO2……."

P12, Fig 6 caption, third line: add "an" so that it reads "an e-folding lifetime"

P12, L243: Investigating -> To investigate

P15, L277: add "a" so that it reads "we also performed a forward simulation……."

P15, L284: termolecular -> thermomolecular

P17, Fig. 10 caption: writing " 1km thick layers between 21 June 2019 and 22 June" is rather confusing since you mix here the vertical layers with the time period you are considering. The altitude range should be given here as well.

P18, L322: SO2 injections separated -> SO2 injections are separated

P18, L329: show -> shows

P18, L333: east of the Raikoke -> east of the Raikoke volcano

P18, 334: either write after the comma "we find that ….." instead of "note that" or swap the two text parts, so that you start the sentence with "Note that……." and continue without comma continuing the sentence with "….comparing……"

P18, L340: by -> on

P19, L347: change "profile de Leeuw" –to "profile shown in de Leeuw" or "shown by"

P19, L348: was over -> was located over

P19, 359: delete "the" so that it reads "of SO2" or add "plume" so that it reads "of the SO2 plume"

P23, Fig. 13 and P24, Fig. 14: Add x-axis title (-> "Date")

P31, L531: replace "the" by "a " so that it reads "considered in a future study".

---

## Author Response (AR2)

**Reply to review comments**

We thank Editor Farahnaz Khosrawi for the time and effort spent on the revised manuscript and for providing detailed technical corrections. We considered all comments and corrected all suggested corrections. Please find our point-by-point replies below (colored in blue).

**Editor Comments**

P1, L2: applied -> apply

Corrected.

P2, L36:"......both day- and nighttime near global coverage" does not sound correct. Please rephrase. Maybe you could write "near global coverage during both, day- and nighttime".

Corrected.

P2, L44: lagranto -> LAGRANTO

Corrected.

P3, L65:"from different studies". This text part does not seem to be at the right place in the sentence. I think you should rather write: For instance the estimation of SO2 mass from the 2009 Sarychev eruption from different studies varies from 0.8 to 1.8 Tg (Fromm et al., 2014)."

Corrected.

P3, L89:"Finally, we discuss the results from our work comparing with previous studies in Sect.4......". Sentence not correct. I would suggest to write it as follows:"Finally, in Sect.4 we discuss the results from our work by comparing to previous studies......"

Corrected.

P4, L94: used -> use

Corrected.

P7, L164: add "a" and "vertical" so that it reads:"...at a 31 km horizontal resolution and 137 vertical levels spanning from the surface up to 0.01 hPa."

Corrected.

P7, L165: interpolated to 0.3 x 0.3 horizotal resolution -> interpolated to a 0.3 x 0.3 horizontal resolution

Corrected.

P7, L177: of the SO2 -> of SO2

Corrected.

P8, L212. Delete "together"

Corrected.

P9, L219: troposphere -> tropospheric

Corrected.

P10, Fig 4 caption: Better to write "plot legend" than "plot key"?

Corrected.

P10, 224: Especially here the usage of retrieval is quite confusing. This paragraph should be reworked.

To clarify, we have used 'product' instead of retrieval here. We have also check throughout the text on the use of 'observation' and 'retrieval', and have changed to used 'product' when we think it is more appropriate.

P11, Fig 5 caption. Add "the' -> "... of the Raikoke SO2......"

Corrected.

P12, Fig 6 caption, third line: add "an" so that it reads "an e-folding lifetime"

Corrected.

P12, L243: Investigating -> To investigate

Corrected.

P15, L277: add "a" so that it reads "we also performed a forward simulation......"

Corrected.

P15, L284: termolecular -> thermomolecular

Corrected.

P17, Fig. 10 caption: writing "1km thick layers between 21 June 2019 and 22 June" is rather confusing since you mix here the vertical layers with the time period you are considering. The altitude range should be given here as well.

Corrected. And the sentence now reads as '...between $0 - 25\,km$ and from 21 June 2019, 18:00 UTC to 22 June 2019, 06:00 UTC.'

P18, L322: SO2 injections separated -> SO2 injections are separated

Corrected.

P18, L329: show -> shows

Corrected.

P18, L333: east of the Raikoke -> east of the Raikoke volcano

Corrected.

P18, 334: either write after the comma "we find that..." instead of "note that" or swap the two text parts, so that you start the sentence with "Note that......" and continue without comma continuing the sentence with "...comparing......"

Corrected.

P18, L340: by -> on

Corrected.

P19, L347: change "profile de Leeuw" to "profile shown in de Leeuw" or "shown by"

Changed to 'profile shown in de Leeuw'.

P19, L348: was over -> was located over

Corrected.

P19, 359: delete "the" so that it reads "of SO2" or add "plume" so that it reads "of the SO2 plume"

Corrected.

P23, Fig. 13 and P24, Fig. 14: Add x-axis title (-> "Date")

Corrected.

P31, L531: replace "the" by "a" so that it reads "considered in a future study".

Corrected.

Additionally, I would like to ask you to check again the occasions where you use "retrieval" in the text. I think simply replacing "observation" by "retrieval" is also not correct. You should for each instance carefully check if the term "observation" or "retrieval" is more appropriate.

We have also check throughout the text on the use of 'observation' and 'retrieval', and have changed back to use 'observation' or 'product' when we think it is more appropriate.